# Unraveling iron oxides as abiotic catalysts of organic phosphorus recycling in soil and sediment matrices

Jade J. Basinski[1], Sharon E. Bone[2], Annaleise R. Klein[1,3], Wiriya Thongsomboon[1,6], Valerie Mitchell[3], John T. Shukle[4,7], Gregory K. Druschel[4], Aaron Thompson [5] & Ludmilla Aristilde [1] ✉

In biogeochemical phosphorus cycling, iron oxide minerals are acknowledged as strong adsorbents of inorganic and organic phosphorus. Dephosphorylation of organic phosphorus is attributed only to biological processes, but iron oxides could also catalyze this reaction. Evidence of this abiotic catalysis has relied on monitoring products in solution, thereby ignoring iron oxides as both catalysts and adsorbents. Here we apply high-resolution mass spectrometry and X-ray absorption spectroscopy to characterize dissolved and particulate phosphorus species, respectively. In soil and sediment samples reacted with ribonucleotides, we uncover the abiotic production of particulate inorganic phosphate associated specifically with iron oxides. Reactions of various organic phosphorus compounds with the different minerals identified in the environmental samples reveal up to twenty-fold greater catalytic reactivities with iron oxides than with silicate and aluminosilicate minerals. Importantly, accounting for inorganic phosphate both in solution and mineral-bound, the dephosphorylation rates of iron oxides were within reported enzymatic rates in soils. Our findings thus imply a missing abiotic axiom for organic phosphorus mineralization in phosphorus cycling.

Phosphorus (P), an essential element for all organisms, has a unique yet incompletely understood biogeochemical cycle. The predominant P form in the critical zone is the oxyanion, phosphate ($PO_4^{3-}$), which cycles between precipitated minerals (e.g. apatite), adsorbed, and dissolved inorganic and organic species in soils[1], sediments[2], freshwater and marine[3,4] environments, and atmospheric dust[5]. A significant fraction of the total P in environmental matrices is found as organic phosphate ($P_{org}$), accounting for 20–70% of total P in soils[6] and 26–64% in lake sediments[7]. Since orthophosphate or inorganic phosphate ($P_i$) is the primary bioavailable form for biological P uptake and assimilation, plants and microbes employ phosphatases and related enzymes to catalyze the hydrolysis of $P_{org}$ to produce $P_i$ through nucleophilic attack[8–11]. This enzymatic reaction has been the focus of studies on $P_{org}$ transformation in environmental matrices. However, there is growing research implicating also mineral surfaces, in particular iron (Fe) oxides and oxyhydroxides, in mediating $P_{org}$ dephosphorylation[12–16], the mechanism of which may resemble the metal active sites of phosphate-cleaving enzymes[17] such as alkaline phosphatase.

The Fe oxide and Fe oxyhydroxide minerals, hereafter referred to collectively as Fe oxides, play a critical role in the regulation of biogeochemical cycles such as P through species adsorption and

[1]Department of Civil and Environmental Engineering, Northwestern University, Evanston, IL, USA. [2]Stanford Synchrotron Radiation Light Source, SLAC National Accelerator Laboratory, Menlo Park, CA, USA. [3]Australian Synchrotron, Australian Nuclear Science and Technology Organisation, Clayton, VIC, Australia. [4]Department of Earth Sciences, Indiana University-Purdue University Indianapolis, Indianapolis, IN, USA. [5]Department of Crop and Soil Sciences, University of Georgia, Athens, GA, USA. [6]Present address: Department of Chemistry, Mahasarakham University, Mahasarakham, Thailand. [7]Present address: ZevRoss Spatial Analysis, Ithaca, NY, USA. ✉e-mail: ludmilla.aristilde@northwestern.edu

subsequent release following desorption or mineral dissolution[3,18,19]. Consequently, Fe oxide adsorbents are considered to serve as both a source and a sink of P in natural environments[20,21]. The adsorption of $P_i$ and $P_{org}$ species onto Fe oxides has been extensively studied[22–27]. The Fe oxide content in soils and sediments, which can be up to 166 g Fe kg$^{-1}$, has been reported to sequester up to half of the total soil P[18]. However, the difficulty of determining which specific P species ($P_i$ or $P_{org}$) are associated with Fe oxides has prevented a comprehensive understanding of the role of Fe oxides in mediating the fate of $P_{org}$.

Prior studies of mineral-mediated hydrolytic cleavage of $P_{org}$ have focused on monitoring products in solution. Using ultraviolet-visible (UV-vis) absorption spectroscopy for solution analysis of para-nitrophenyl phosphate ($p$-NPP)[12], a synthetic model $P_{org}$, Fe oxides (goethite, hematite, and an amorphous phase), manganese (Mn) oxides (akhtenskite, pyrolusite, and an amorphous-type phase), and titanium oxides (anatase and rutile) were all shown to be effective at cleaving the phosphoester bond in this synthetic $P_{org}$. With respect to environmentally relevant $P_{org}$, Fe oxide nanoparticles were reported to mediate the dephosphorylation of adenosine-5′-triphosphate (ATP, a triphosphorylated ribonucleotide) and glucose-6-phosphate (G6P, a monophosphorylated sugar) based on colorimetric measurements of $P_i$ in solution using molybdate and UV-vis absorbance[13]. Combining this colorimetric technique with $^{31}$P nuclear magnetic resonance for solution $P_i$ analyses, a crystalline Fe oxide (hematite), a hydrated Mn oxide (birnessite), and, to a lesser degree, an aluminum (Al) oxide (boehmite) were all reported to generate $P_i$ in solution from the phosphomonoesters G6P, glycerophosphate, and adenosine-5′-monophosphate (AMP), as well as the triphosphorylated ATP[16]. High-resolution liquid chromatography-mass spectrometry (LC-MS) has been used for solution analysis of organic products to confirm the dephosphorylation of ATP, adenosine-5′-diphosphate (ADP), and AMP by ferrihydrite, a short range-ordered Fe oxide[14] and birnessite[28]. Furthermore, hydrolysis of RNA by goethite and hematite was reported by using high-performance LC and UV-vis absorbance to detect nucleobases in solution[29]. Therefore, the traditional view that adsorbed $P_{org}$ on Fe oxides is protected from transformation due to a presumed low catalytic reactivity of Fe oxides needs to be revisited, particularly within the context of P cycling.

Yet, the aforementioned solution-based data captured an incomplete picture of the mineral-catalyzed $P_{org}$ dephosphorylation, because adsorbed (or particulate) species associated with the mineral surface could include the produced $P_i$ from $P_{org}$. Advances in mineral surface characterization by synchrotron-based P K-edge X-ray absorption near-edge structure (XANES) spectroscopy have made it possible to distinguish between $P_i$ and $P_{org}$ bound to Fe in minerals[30] or Fe oxides in a soil matrix[31]. In terms of monitoring mineral-catalyzed $P_{org}$ dephosphorylation, one application of the XANES technique with ferrihydrite revealed the generation of particulate $P_i$ from adsorbed ribonucleotides, while $P_i$ was notably absent from solution[14]. This latter finding, which was confirmed by quantifying the dephosphorylated organic products in solution by LC-MS[14], highlights the need for quantitative analysis of particulate $P_i$, in addition to dissolved $P_i$, especially for minerals such as Fe oxides with strong adsorption affinity for $P_i$ species. For instance, soil P content was found to be associated predominantly with semi-crystalline Fe oxides[18], but it remains unknown whether these minerals contribute to the particulate $P_i$ fraction in soil by catalyzing the dephosphorylation of mineral-bound $P_{org}$. Furthermore, Fe oxide-associated $P_{org}$ fractions have been implicated in enhancing plant-bioavailable P, but an underlying abiotic catalytic process has not been explored[32].

Here we tested the hypothesis that Fe oxides would catalyze the recycling of $P_i$ through the hydrolytic cleavage of $P_{org}$ bound to Fe oxides in heterogeneous environmental matrices. To this end, we coupled the surface-sensitive XANES analysis with high-resolution LC-MS to investigate the role of Fe oxides as simultaneous adsorbents and catalysts for ribonucleotides reacted with natural sediment and soil samples. As remnants of nucleic acids, ribonucleotides represent a ubiquitous and abundant class of naturally occurring $P_{org}$ compounds. A major fraction of $P_{org}$ is stored in RNA and DNA, which have been found in soils at 56 µg g$^{-1}$ and 435 µg g$^{-1}$, respectively[33,34]. Through biotic and abiotic reactions, monophosphorylated ribonucleotides are generated from the nucleic acid polymers; moreover, diphosphorylated and triphosphorylated ribonucleotides are essential metabolites widely involved in carbon and energy metabolism in plants and microorganisms[3], and may account for > 25% of P content in certain microbial populations[4,35]. In this study, we focused on performing reactions with two ribonucleotides with different phosphate bonding: ATP (containing a phosphoester and two phosphoanhydride, P-O-P, bonds) and AMP (containing only a phosphoester, C-O-P, bond)[9]. Because phosphomonoesters are common $P_{org}$ types in sediment and soil systems[36–38], we performed experiments with two other relevant naturally-occurring phosphomonoesters: G6P, a sugar phosphate involved in sugar metabolism; inositol hexakisphosphate or phytate, the primary P storage in terrestrial plants. To investigate the reactivity of the different mineral types in the natural samples, we reacted the four different $P_{org}$ compounds with pure forms of Fe oxides (goethite, hematite, and ferrihydrite) and various silicate-bearing minerals (quartz, clays, mica). Taken collectively, the findings from this research challenge the lack of consideration in the P cycle, beyond enzymatic processes, of an abiotic contribution of Fe oxide minerals in the generation of $P_i$ from particulate $P_{org}$ sources.

## Results
### Characterization of the sediment and soil samples
For our experiments with natural samples, we chose lake sediment and forest soil samples with similar endogenous Fe content (~50 mg Fe kg$^{-1}$ dry sample), excavated from Missisquoi Bay in the U.S. state of Vermont and an Ultisol from the Calhoun Experimental Forest in the U.S. state of South Carolina, respectively (Fig. 1a). Missisquoi Bay is a eutrophic section of Lake Champlain that undergoes diel and seasonal oxic-anoxic cycles[39] (Fig. 1a, b). These redox cycles promote dynamic formation of Fe oxides, which are implicated in the mobility and fate of the bioavailable sediment P, albeit the mechanism is not well understood. In this lake environment, P transformation is attributed to enzymatic and biological activity, but little is known about the potential conversion of $P_{org}$ to $P_i$ by Fe oxides (Fig. 1b). In our complimentary forest setting, the soils at the Calhoun Experimental Forest contain redoximorphic features and undergo redox fluctuations in response to rainfall and organic carbon pulses, leading to periodic dissolution and reformation of Fe minerals[40] (Fig. 1b).

We combined multiple characterization techniques to determine the specific minerals in the sediment and soil samples that would be responsible for adsorption and catalytic reactivities towards $P_{org}$ compounds: X-ray fluorescence (XRF) to determine the major elements present [Supplemental Information (SI), Fig. S1]; X-ray diffraction (XRD) to identify the major crystalline mineral phases (quartz, micas, feldspars, clays, or Fe oxides) in accordance with the elemental composition from the XRF analysis (SI, Fig. S2); and, Fe K-edge XANES spectroscopy to resolve different forms of Fe phases such as Fe oxides of low crystallinity (i.e., ferrihydrite) and high crystallinity (i.e., hematite, goethite), an Fe phosphate mineral (i.e., vivianite), and an Fe-rich mica (i.e., biotite) (SI, Fig. S3-Fig. S5, and Table S1). While our Fe XANES data accounted for low-crystallinity Fe-mineral phases, our XRD data do not account for the possible presence of low-crystallinity silicate or aluminosilicate phases in the natural samples. With respect to the possible presence of low-crystallinity aluminum (Al)-containing minerals, we obtained $^{27}$Al nuclear magnetic resonance (NMR) spectra of the sediment and soil samples to probe for the expected five-coordinated Al in the reactive sites of these minerals[41–43]. Our NMR data only detected tetrahedral and octahedral Al as would be found in clays

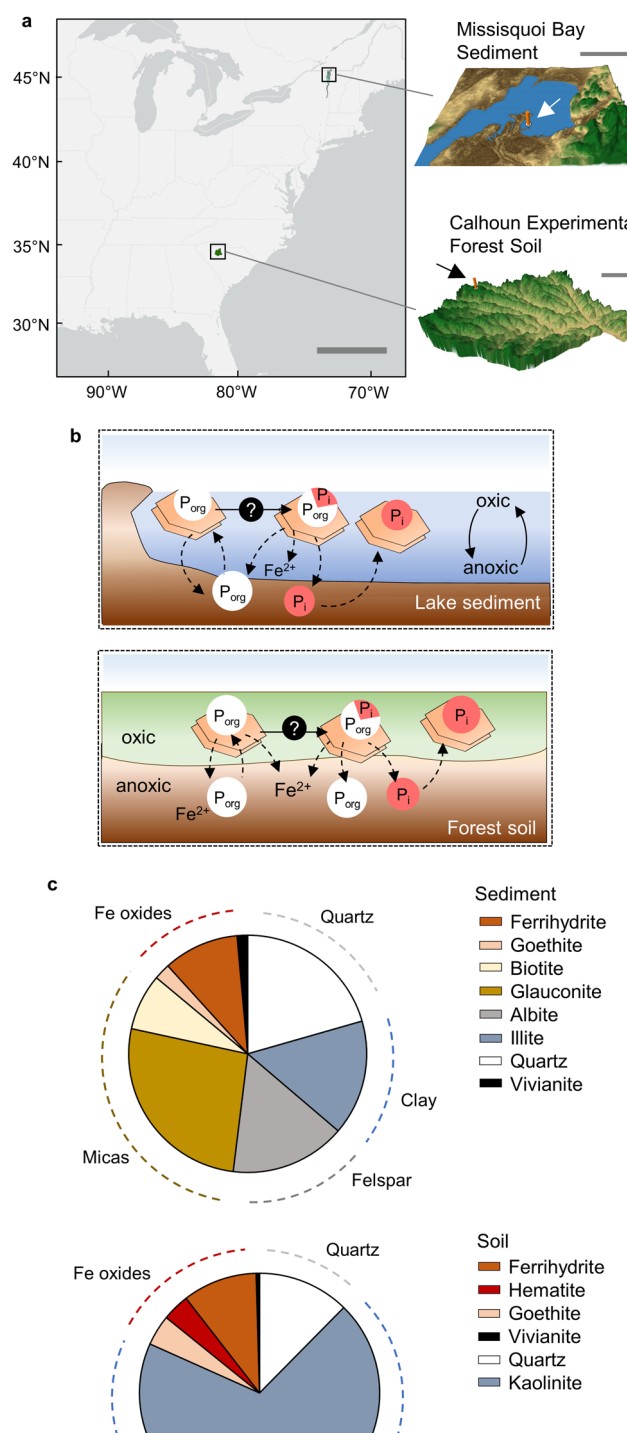

**Fig. 1 | Characterization of the sediment and soil samples. a** (left) Location and (right) topography of (top) Missisquoi Bay (N44°59′33″ W73°8′20″) and (bottom) Calhoun Critical Zone Observatory (N34°36′33.012″ W81°43′40.62″) for lake sediment and forest soil sampling sites, respectively. Scale bar in (**a**) represents 500 km in geographical location and 10 km in topography visualization; arrow and orange vertical line specify sampling location. Maps were created using QGIS and topography data from the U.S. Geological Survey. **b** Illustration of the current view for the abiotic fate of $P_{org}$ in (top) lake and (bottom) soil; legend for (**b**): organic phosphorus ($P_{org}$; white circle), inorganic phosphorus ($P_i$; light red circle), dissolved iron (Fe) ($Fe^{2+}$; black text), Fe oxide mineral (light brown hexagon). **c** Mineral content in the (top) sediment and (bottom) soil samples: ferrihydrite (brown), hematite (red), goethite (peach), biotite (yellow), glauconite (gold), albite (gray), quartz (white), and illite or kaolinite (blueish gray). In (**c**) mineral composition was determined by combining X-ray fluorescence, X-ray diffraction, and Fe X-ray absorption near-edge structure spectroscopy data (SI, Fig. S1-S5). Source data are provided as a Source Data file.

about 12% of the sediment contained Fe oxides identified as ferrihydrite (~10%) and goethite (2%), with a minor fraction (<1%) present as vivianite (Fig. 1c). The soil sample contained primarily kaolinite (a 1:1 clay, 69%) and quartz (12%), with the remaining fraction constituted of Fe oxides as ferrihydrite (10%), hematite (4%), and goethite (4%); vivianite was also present as a minor fraction (<1%) (Fig. 1c). Our sediment characterization was in agreement with a previously reported mineral composition for the sediment sample[44], albeit the Fe content was greater than a previously reported value[39]. For the soil, the ratios of Fe oxide minerals in the soil mineral composition were in close agreement with the reported values from extracted Fe content, but the absolute values were not directly comparable to previous [57]Fe Mössbauer spectroscopy data[45]. These discrepancies with previous data were likely due to differences in sample preparation or characterization techniques, such as the different scales of ordered minerals probed by Fe XANES versus Mössbauer spectroscopies and characterization of separate 5 μm soil sections versus bulk sample. In sum, based on our analysis, we found that about 80% (or more) on a per-mass basis of both sediment and soil samples was comprised of silicate minerals of different types (quartz, micas, feldspars, clays), and <20% constituted the Fe oxide fraction (Fig. 1c). Next, we explored the reactivities of these different mineral types in the environmental samples towards $P_{org}$ compounds.

## Exceptional adsorption and catalysis by Fe oxides

We performed reactions of various $P_{org}$ compounds with the different mineral types identified in the natural samples (at 1 g L$^{-1}$ or 4 g L$^{-1}$): biotite (a mica), quartz, kaolinite (a 1:1 clay), illite (a 2:1 clay), ferrihydrite (a low-crystallinity Fe oxide), hematite (a crystalline Fe oxide), and goethite (a crystalline Fe oxide) (Fig. 2; SI, Fig. S6–Fig. S10). In our first set of experiments, we chose ATP (ATP-P, 150 μM or 4.6 mg L$^{-1}$) as our representative $P_{org}$ that contains both phosphoanhydride (P-O-P) and phosphoester (C-O-P) bonds that are ubiquitous in the reservoir of $P_{org}$ compounds derived from metabolism and biomass of plants and microbes. We used high-resolution LC-MS to quantify particulate and aqueous species of $P_i$ and $P_{org}$ for the ribonucleotide reactants and products in solution (i.e., ATP, ADP, AMP, adenosine), a visible light absorption spectroscopy method for solution $P_i$, and synchrotron-based P K-edge XANES spectroscopy for the relative fractions of particulate $P_i$ and particulate $P_{org}$ associated with Fe-bearing minerals; particulate $P_i$ on non-Fe minerals was determined via mass balance based on the concentrations of organic and inorganic products in solution after reactions with a $P_{org}$ reactant. We note that the XANES analysis can determine the relative organic fraction versus inorganic fraction of the total particulate P, but does not discriminate between possible different $P_{org}$ compounds in the particulate $P_{org}$ fraction[14].

Following the reaction with ATP, quartz and the aluminosilicates (mica, kaolinite, illite) exhibited minimal to no catalytic reactivity whereby aqueous $P_i$ or particulate $P_i$ accounted for <4% of the reacted

and micas, which can be identified by XRD; no five-coordinated Al was detected (SI, Fig. S6). Furthermore, as will be discussed in the next section, there was minimal to no catalytic reactivity of silicate or aluminosilicates towards the different $P_{org}$ compounds (Fig. 2). No Mn-bearing minerals were included in the mineral composition analysis because neither environmental sample exhibited a quantifiable amount of Mn (SI, Fig. S1).

Taken collectively, our analysis revealed that the sediment sample contained primarily glauconite (a mica, 26%), followed by quartz (21%), illite (a 2:1 clay, 16%), albite (a feldspar, 16%), and biotite (a mica, 8%);

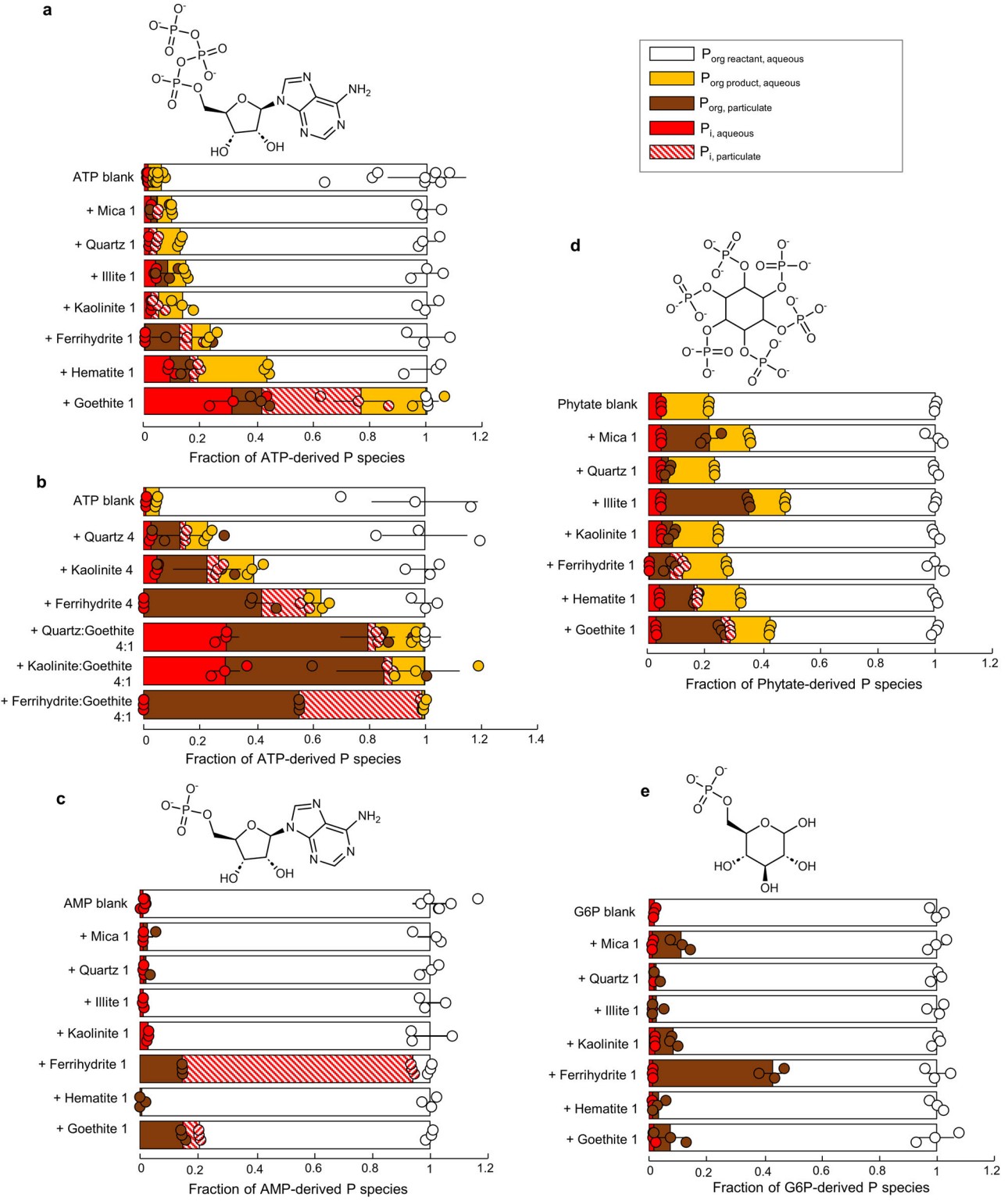

**Fig. 2 | Catalytic and adsorption reactivities of minerals towards different organic phosphorus ($P_{org}$) structures. a, b** Transformation of adenosine triphosphate (ATP)-P (150 μM or 4.6 mg L$^{-1}$) after 7 day reaction with mica, illite, quartz, kaolinite, ferrihydrite, hematite, or goethite at either 1 g L$^{-1}$ (e.g., Quartz 1) or 4 g L$^{-1}$ (e.g., Quartz 4). Transformation of (**c**) adenosine monophosphate (AMP)-P (50 μM or 1.5 mg L$^{-1}$), (**d**) phytate-P (300 μM or 9.0 mg L$^{-1}$), and (**e**) glucose-6-phosphate (G6P)-P (50 μM or 1.5 mg L$^{-1}$) after 7 day reaction with mica, quartz, illite, kaolinite, ferrihydrite, hematite, or goethite (1 g L$^{-1}$). Data are represented as mean values +/-

SD where $n$ = 3 independent samples except for $n$ = 9 for ATP blank and $n$ = 6 for AMP blank. Aqueous inorganic P ($P_i$) and $P_{org}$ species were determined by ultraviolet-visible absorption spectroscopy and high-resolution liquid chromatography mass spectrometry analysis, respectively; particulate $P_i$ and $P_{org}$ species were determined by P K-edge X-ray absorption near-edge structure spectroscopy. Color codes: dissolved $P_{org}$ reactant (white), dissolved $P_{org}$ product (yellow), particulate $P_{org}$ (brown), dissolved $P_i$ (red), particulate $P_i$ (red stripes). Source data are provided as a Source Data file.

$P_{org}$; in terms of adsorption reactivity, less than 20% of the reacted $P_{org}$ was found as particulate $P_{org}$ with these minerals (Fig. 2a). In contrast, the reactions with the different Fe oxides (ferrihydrite, hematite, or goethite) yielded substantial dephosphorylation or adsorption of the $P_{org}$ reactant, characterized by up to 31% as aqueous $P_i$, 12−36% as particulate $P_i$, and up to 46% as particulate $P_{org}$ (Fig. 2a). In regards to the fate of the reacted P (i.e., sum of transformed and adsorbed $P_{org}$-derived P) across the Fe oxide minerals, ferrihydrite exhibited the highest adsorption with greater than 75% of reacted P as particulate $P_{org}$, while goethite was the most catalytically active, with 53−79% of the reacted P as collectively adsorbed $P_i$ and aqueous $P_i$ (Fig. 2a). Despite a greater than 10-fold difference in the specific surface area of ferrihydrite ($230\,m^2\,g^{-1}$) compared to goethite ($16\,m^2\,g^{-1}$), there was a near 3-fold higher fraction of particulate P species on goethite than on ferrihydrite ($p < 0.05$) (Fig. 2a), suggesting that mineral surface chemistry rather than specific surface area governs the extent of catalytic reactivity, a worthwhile research avenue to pursue in future investigations.

### Catalytic reactivity of Fe oxides in mineral mixtures

Since Fe oxides are present in heterogeneous mineral mixtures in natural samples, we also investigated how the presence of other minerals (ferrihydrite, quartz, or kaolinite) would influence the high catalytic reactivity of goethite towards ATP (Fig. 2b). We prepared mineral mixtures based on the composition of the natural samples (Fig. 1c). First, with a 4:1 ferrihydrite:goethite mixture, we found that the amount of particulate $P_i$ generated in the mixture was equivalent to that of goethite alone ($p = 0.21$) while there were an 8-fold increase in particulate $P_{org}$ ($p < 0.001$) and no $P_i$ in solution, indicating that the higher adsorption reactivity of ferrihydrite relative to goethite overwhelmed the reactivity in the ferrihydrite:goethite mixture (Fig. 2b). Second, when comparing the goethite-only condition to 4:1 quartz:goethite and kaolinite:goethite mixtures, the same amount of dissolved $P_i$ was produced from the $P_{org}$ reactant ($p \geq 0.75$) but only 20% of the particulate $P_i$ fraction remained ($p < 0.01$), and this was accompanied by a 6-fold increase in particulate $P_{org}$ ($p < 0.01$) (Fig. 2b). Our data thus implied that, in terms of the production of solution $P_i$ or particulate $P_i$, Fe oxides would likely retain their catalytic reactivity in mixed-mineral matrices in environmental samples.

### Catalytic reactivity of Fe oxides with different $P_{org}$ types

Amongst the chemical diversity of $P_{org}$ types found in biomolecules, phosphomonoesters are widely found in soils[36–38,46,47]. To probe Fe oxide reactivity with these other types of $P_{org}$, we performed reactions involving each of the three Fe oxides (ferrihydrite, hematite, and goethite) with three phosphomonoesters: AMP (AMP-P, 50 $\mu$M P or 1.5 mg P $L^{-1}$), G6P (G6P-P, 50 $\mu$M P or 1.5 mg P $L^{-1}$); and phytate (phytate-P, 300 $\mu$M P or 9.0 mg P $L^{-1}$) (Fig. 2c−e). To compare the Fe oxide reactivity with the other mineral types in the environmental samples, we also performed experiments of the phosphomonoester compounds reacted with quartz and the aluminosilicates (mica, kaolinite, illite) (Fig. 2c−e).

In contrast to the ATP reactions, all the $P_i$ derived from the reacted AMP with goethite and ferrihydrite was retained as particulate $P_i$ while aqueous $P_i$ was absent, a significant finding that was made possible here due to the application of the XANES technique (Fig. 2c). On the one hand, the catalytic reactivity of ferrihydrite was higher for AMP than for ATP, as reflected by the 20-fold increase in the particulate $P_i$ fraction ($p < 0.001$) (Fig. 2a, c). On the other hand, the catalytic reactivity of goethite was less for AMP than for ATP as characterized by a 3-fold decrease in particulate $P_i$ fraction ($p < 0.01$) accompanied by no change in the particulate $P_{org}$ fraction ($p = 0.29$) and no measured solution $P_i$ (Fig. 2a, c). Hematite did not display any adsorption or catalytic reactivity towards AMP (Fig. 2a and c). These findings here are consistent with a previous report[15] of 2-fold to 6-fold greater reactivity of goethite than hematite for $p$-NPP dephosphorylation, albeit direct comparison between $p$-NPP (a synthetic $P_{org}$) and AMP (a natural biomolecule) is not appropriate due to the smaller molecular weight of $p$-NPP versus AMP ($263.1\,g\,mol^{-1}$ versus $324.23\,g\,mol^{-1}$) and the difference in their chemical structures such as the presence of a sugar base and two heterocyclic nitrogenous rings in AMP whereas $p$-NPP has one benzyl ring. Interestingly, after all the ATP reacted with goethite was transformed or adsorbed in our aforementioned experiments with ATP and goethite, 20−24% of the initial ATP-P remained as AMP-P in solution and no ADP was detected (SI, Fig. S11). The results with the AMP-goethite experiment implied that this accumulation of AMP in the ATP-goethite experiment was due to the lower reactivity of goethite for AMP relative to ATP (Fig. 2c; SI, Fig. S11). Notably, the silicate and aluminosilicate minerals did not exhibit any adsorption or catalytic reactivity towards AMP (Fig. 2c).

With respect to phytate, the silicate and aluminosilicate minerals were all found to adsorb phytate (from 5% to 50% of the total reacted phytate-P) and, rather than an Fe oxide, the clay illite adsorbed the most phytate (Fig. 2d). However, similar to the results with AMP and ATP, catalytic reactivity towards phytate was only obtained with Fe oxides (Fig. 2d). Relative to controls, phytate-derived particulate $P_i$ was higher by 8−14% ($p < 0.01$) with hematite and by 24−32% with goethite ($p < 0.001$); the particulate $P_i$ with ferrihydrite, however, corresponded to the adsorption of solution $P_i$ from the control experiment (Fig. 2d). As with AMP, the most significant adsorption for G6P was with ferrihydrite, with 47−78% of the reacted G6P found only as particulate $P_{org}$ (Fig. 2e). Some G6P adsorption (4−12%) was observed with two of the aluminosilicates (mica and kaolinite) and one of the other Fe oxides (goethite) (Fig. 2e). None of the investigated minerals catalyzed G6P dephosphorylation (Fig. 2e). Only data with ferrihydrite and goethite were consistent with reports by Wan et al.[16] of higher aqueous $P_i$ production from AMP than from G6P and phytate. Even accounting for particulate $P_i$, we obtained lower reactivity of AMP and G6P with hematite than reported by Wan et al.[16]. Compared to our experiments, these previous experiments[16] were performed with 60% less hematite (on a per-mass basis), 20-fold higher concentration for G6P and AMP, and 3-fold higher concentration for phytate (Fig. 2c−e). Therefore, the discrepancy could be due to possible lower reactive sites in hematite compared to ferrihydrite and goethite, thus necessitating higher $P_{org}$ concentration to observe reactivity.

In sum, our data revealed that the silicates and aluminosilicates either had minimal to no catalytic reactivity or exhibited variable extent of adsorption reactivity towards the different compounds. Only the Fe oxides were found to catalyze appreciable to significant dephosphorylation of $P_{org}$ compounds containing phosphoester and phosphoanhydride bonds. Importantly, the Fe oxide-catalyzed reactions seemed to be dependent on both the mineral surface chemistry and the type of $P_{org}$ species. As proposed previously[14,28], we posit that the differences in catalytic reactivity may stem from the binding conformations of different $P_{org}$ compounds on the surface of the Fe oxide minerals.

### Evolution of Particulate $P_i$ from reacted $P_{org}$ in natural samples

Our findings with the pure minerals stressed the exceptional adsorption and catalytic reactivities of the Fe oxides relative to the other minerals identified in the environmental samples. Notably, we found that particulate $P_i$, which was largely ignored in previous investigations, was an important product of the reaction of Fe oxides with $P_{org}$ species bearing a phosphomonoester bond or a phosphoanhydride bond. Furthermore, our data with mineral mixtures implied that the Fe oxide-mediated catalysis would likely remain prominent even in the heterogeneous mineral matrix of environmental samples. To evaluate this, we performed $P_{org}$ reactions with natural sediment and soil samples.

First, we determined the starting P content in the natural samples: the sediment sample contained, on average, 1334 mg P per kg, with 49% as particulate $P_{org}$ and 51% as particulate $P_i$; the soil sample

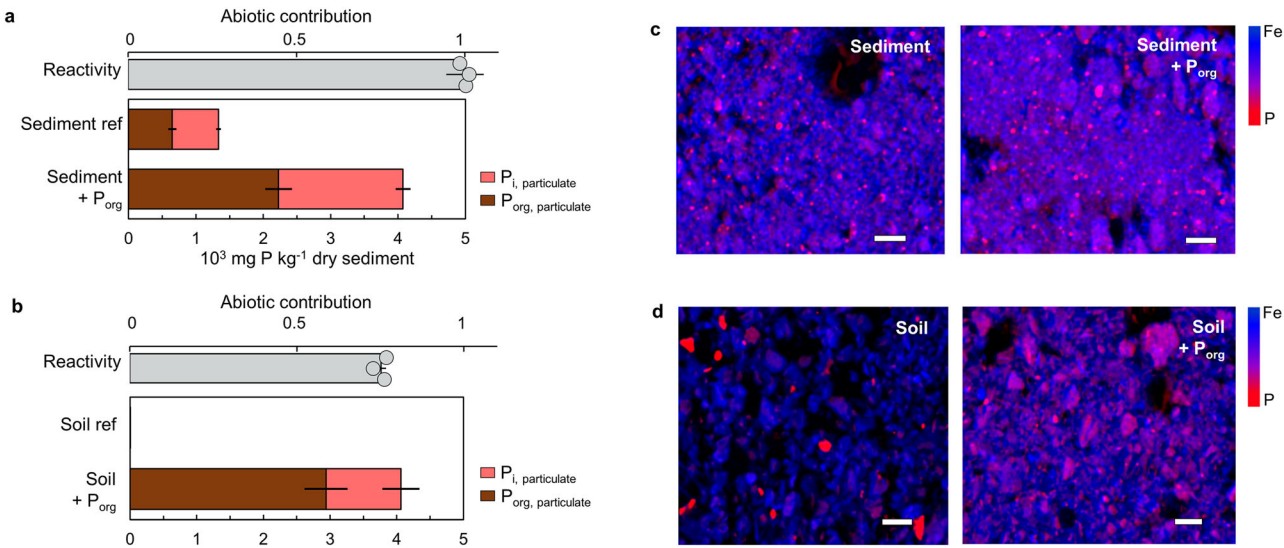

**Fig. 3 | Particulate inorganic phosphorus ($P_i$) generated from ribonucleotides reacted with natural samples.** For (**a**) sediment and (**b**) soil samples: (top) Contribution of abiotic reactivity (gray) and (bottom) distribution of particulate organic phosphorus ($P_{org}$; brown) and particulate $P_i$ (pink) before (reference, ref) and after 7 d reaction with $P_{org}$ as adenosine triphosphate (ATP) (ATP-P, 300 μM or 9.3 mg $L^{-1}$) with 1 g $L^{-1}$ of the dry sediment or soil sample. Error bars represent (top) standard deviation of 3 independent replicates for abiotic contribution or (bottom) errors in X-ray absorption near-edge structure spectroscopy model fitting for particulate P speciation. **c**, **d** μ-X-ray fluorescence mapping of count intensity for iron (Fe) (blue, max intensity = 500 for **c** or 900 for **d**) and P (red, max intensity = 25,000) in (**c**) sediment or (**d**) soil before (left) and after (right) 7 d reaction with ATP. In (**a**) and (**b**) background particulate P species in the reference sediment and soil samples are noted as "Sediment ref" and "Soil ref", respectively. In (**c**) and (**d**) the scale bars (shown in white) represent 300 μm. Source data are provided as a Source Data file.

only had 0.43 mg P per kg, with 56% as particulate $P_{org}$ and 44% as particulate $P_i$ (Fig. 3a). Relative to the starting P content in the natural samples, reacting ATP-P (300 μM or 9.3 mg $L^{-1}$) with the sediment and soil samples (at 1 g $L^{-1}$) generated excess particulate P of, on average, 2.7 mg P per g of the sediment sample and 4.1 mg P per g of the soil sample (Fig. 3a, b). Importantly, of this ribonucleotide-sourced particulate P, 20 – 50% was particulate $P_i$ and 52 – 79% was particulate $P_{org}$ (Fig. 3a, b). Therefore, following reactions with ATP, the sediment sample had a near 3-fold increase in the amount of both $P_{org}$ and $P_i$ in the particulate fraction and the soil sample had 3 orders of magnitude higher of particulate $P_{org}$ and $P_i$ (Fig. 3a). Interestingly, of the total $P_i$ evolved from the $P_{org}$ reactant, only about one-third was solution $P_i$ while nearly two-thirds remained as particulate $P_i$ fraction, thus highlighting a significant underestimation of the produced $P_i$ would result in the absence of particulate P speciation (SI, Fig. S12). Biologically mediated $P_{org}$ dephosphorylation was not expected to be significant in our natural samples due to long-term storage (~4 years) of both samples and low carbon loading (<0.2% g C $g^{-1}$ soil) particularly for the soil sample. Nevertheless, we tested the possibility of residual microbial or enzymatic reactions in the natural samples by performing experiments with an antimicrobial agent or an enzyme denaturing agent, respectively (SI, Fig. S13). We determined that these biotic reactions accounted for only 0–5% and 23 – 26% of the total reactivity in the sediment and soil samples, respectively (Fig. 3a, b). Taken collectively, our findings bring attention to the occurrence of a pool of abiotically generated particulate $P_i$ from mineral-mediated $P_{org}$ transformation that has been hitherto unaccounted for in environmental matrices. Next, we probed which of the minerals within the heterogeneous matrix of the natural samples may be responsible for the $P_{org}$ reactivity.

### Association of evolved $P_i$ with Fe oxides in natural samples
Through μ-XRF mapping, we found that Fe was distributed throughout the sediment and soil samples and that P appeared to be co-localized with Fe in distinct regions (Fig. 3c and d); we confirmed the lack of strong correlation of P with calcium (Ca) (SI, Fig. S14). Subsequently,

we obtained extensive bulk P K-edge XANES data to determine the mineral types associated with the particulate P generated from the reacted ribonucleotide by conducting linear combination fitting (LCF) using XANES spectra obtained from separate experiments of $P_{org}$ or $P_i$ with the different mineral components identified in the natural samples (Fig. 4; SI, Table S6 and Table S7). It was not possible to use the XANES spectra to distinguish the specific P species associated with the different aluminosilicates nor the specific Fe oxide minerals associated with either particulate $P_i$ or particulate $P_{org}$ species (SI, Fig. S15). Specifically, we were able to employ the LCFs of the XANES spectra to distinguish $P_i$ associated with Ca using apatite as a reference, P (without discriminating between $P_i$ or $P_{org}$) associated with silicates and aluminosilicates, $P_i$ associated with Fe in Fe oxides, $P_{org}$ species associated with Fe in Fe oxides, and $P_i$ in Fe-$P_i$ clusters using vivianite as reference (Fig. 4a–e). We determined the fate of newly generated $P_i$ and $P_{org}$ following the ATP reaction with the natural samples by accounting for the total particulate P in excess of the background amount in the reference samples and characterizing the differences in the fractions of $P_i$ and $P_{org}$ associated with specific minerals in natural samples before and after reactions with ATP (Fig. 4f, g). The following were characteristic features in the XANES spectra that enabled the LCFs to capture distinct P speciation in the natural samples reacted with $P_{org}$: differences in the rising edge of the white line for $P_{org}$ versus $P_i$ (Fig. 4c, d), the shift in the peak of the white line with different fractions of $P_i$ versus $P_{org}$ bound to Fe oxides (Fig. 4d), and the pre-edge region of silicate-bound P (4e).

In accordance with the observed higher reactivity of the Fe oxides relative to the other minerals in the reactions with pure minerals (Fig. 2a, b), we found that both particulate $P_{org}$ and particulate $P_i$ generated after reactions of the sediment and soil samples with the ribonucleotide reactant were associated primarily with Fe and Fe-oxide fractions in the sample matrix (Fig. 4f, g). On a dry sediment basis, about 72% of the ribonucleotide-derived particulate P was found in clusters of Fe-complexed $P_i$ (1066 ± 118 μg $g^{-1}$) and particulate $P_{org}$ associated with the Fe oxide fraction (912 ± 502 μg $g^{-1}$); the remaining

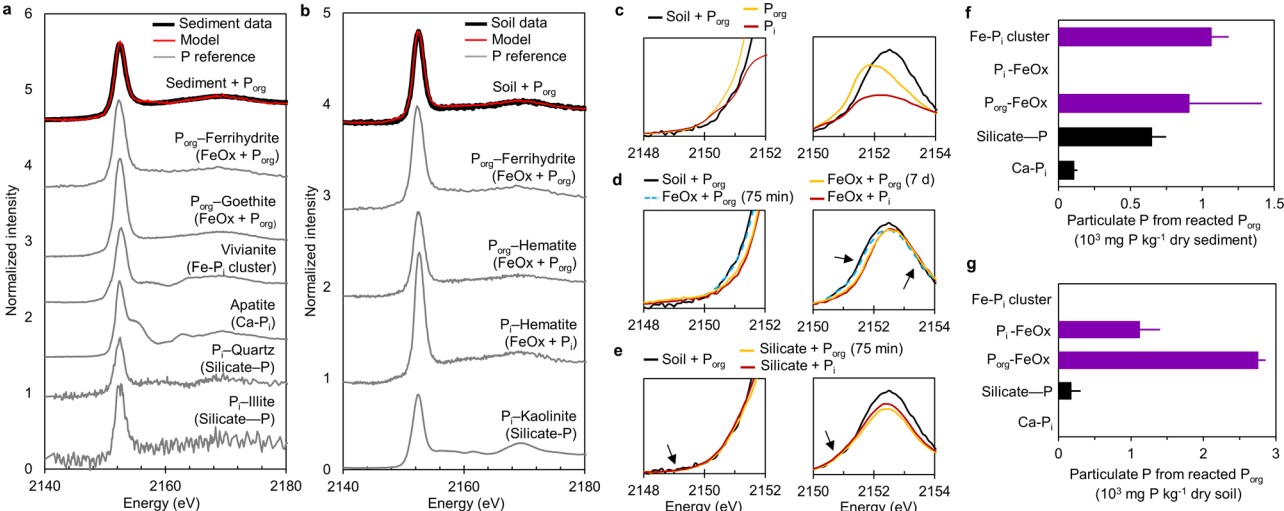

**Fig. 4 | Specific association of organic phosphorus ($P_{org}$)-derived inorganic phosphorus ($P_i$) with iron (Fe) and Fe oxides in sediment and soil samples. a, b** Bulk P K-edge X-ray absorption near-edge structure spectroscopy (XANES) data (black line) of the (**a**) sediment and (**b**) soil samples after 7 d reaction with the $P_{org}$ reactant (adenosine triphosphate; ATP); the model fits (red line) are from linear combination fitting (LCF) of reference spectra (gray lines) of $P_i$ or $P_{org}$ reacted with the different mineral types identified in the samples (see Fig. 1c). **c, d, e** Close-up of spectral regions of the XANES data obtained with: (**c**) the reference $P_{org}$ (ATP, prepared as ATP disodium salt hydrate) powder, the reference $P_i$ (prepared as disodium phosphate), and the soil reacted with $P_{org}$ reactant; (**d**) a representative Fe oxide (FeOx) mineral (goethite) reacted with ATP ($P_{org}$) for 75 min or 7 days, the same mineral reacted with $P_i$ for 7 days, and the soil reacted with the $P_{org}$ reactant;

(**e**) a representative aluminosilicate (Al-silicate) mineral (kaolinite) reacted with ATP ($P_{org}$) for 75 min or $P_i$ for 7 days, and the soil reacted with the $P_{org}$ reactant. Color key: soil reacted with ATP (black line); $P_{org}$ reference or $P_{org}$ bound to a mineral (yellow line); $P_i$ reference or $P_i$ bound to a mineral (red line); and $P_{org}$ bound to an Fe oxide (light blue dashed line). **f, g** Speciation of the excess particulate P content ($10^3$ mg P kg$^{-1}$ dry sample) generated following 7 d ATP reaction with (**f**) the lake sediment sample or (**g**) the forest soil sample. Data in (**a**), (**b**), (**f**) and (**g**) are detailed in SI Table S2, Table S3, Table S4, Table S5, Table S6, and Table S7. In (**f**) and (**g**) error bars represent model fitting error range from the LCF of XANES spectra from a single bulk P K-edge XANES measurement. Source data are provided as a Source Data file.

was P associated with aluminosilicate minerals ($651 \pm 97$ µg g$^{-1}$), and $P_i$ in Ca-phosphate mineral ($110 \pm 22$ µg g$^{-1}$) (Fig. 4f). Based on our experiments with pure minerals (Fig. 2a), the accumulation of Fe-$P_i$ clusters in the sediment sample matrix was attributed to strong Fe binding of $P_i$ catalytically generated from the high abundance of particulate $P_{org}$ associated with Fe oxides (Fig. 4f). In contrast to the lake sediment, the particulate P generated from the reacted ribonucleotide with the soil sample was nearly completely associated with the Fe oxide fraction, whereby ~28% was Fe oxide-bound $P_i$ ($1124 \pm 279$ µg g$^{-1}$) and the remaining 72% was Fe oxide-bound $P_{org}$ ($2758 \pm 103$ µg g$^{-1}$); there was a relatively minor amount of P associated with silicates ($181 \pm 127$ µg g$^{-1}$) (Fig. 4g). Based on the different mineral compositions of sediment versus soil samples and informed by our experiments with pure minerals (Fig. 2a), we posit that the observed differences between the natural samples in the mineral associations of the ribonucleotide-sourced P was due to the different forms of Fe oxides (hematite-free versus hematite-rich Fe-oxide fractions), silicate types (quartz-enriched versus clay-enriched), and available Ca (nearly 12-fold higher Ca content in the sediment sample) (Table S8).

## Rates of Fe oxide-mediated catalysis versus soil enzymes
Our data thus far point to the production of both aqueous $P_i$ and particulate $P_i$ during Fe oxide-mediated $P_{org}$ dephosphorylation, the environmental relevance of which needs to be considered in relation to reported enzymatic rates (Fig. 5; SI, S11 and S12). To this end, with the three different Fe oxides (goethite, hematite, ferrihydrite) identified in the natural samples, we performed kinetic experiments to obtain the production rates of aqueous $P_i$ and particulate $P_i$ species as a function of reacted ATP concentrations (25–400 µM), and subsequently determined two apparent maximal dephosphorylation rates ($V_{max}$): one $V_{max}$ for the production rate of dissolved $P_i$ and one $V_{max}$ for the production rate of particulate $P_i$ (Fig. 5a, b; SI, Fig. S16). While the goethite-catalyzed reaction generated both dissolved and particulate

$P_i$, only particulate $P_i$ was found in appreciable quantities with both hematite and ferrihydrite (Fig. 5b). The $V_{max}$ for particulate $P_i$ with ferrihydrite (2.59–4.24 µmol $P_i$ h$^{-1}$ g$_{mineral}^{-1}$) was up to 4-fold higher than with goethite (0.455–1.70 µmol $P_i$ h$^{-1}$ g$_{mineral}^{-1}$, $p < 0.05$) and up to 20-fold higher than with hematite (0.164–0.576 µmol $P_i$ h$^{-1}$ g$_{mineral}^{-1}$, $p < 0.05$) (Fig. 5b). This marked difference in the production rate of particulate $P_i$ is consistent with the well-known higher adsorption reactivity of low-crystallinity Fe oxides relative to crystalline Fe oxides[48–50]. With goethite, the $V_{max}$ for $P_i$ in solution (2.70 – 4.65 µmol $P_i$ h$^{-1}$ g$_{mineral}^{-1}$) was nearly 5-fold greater than the corresponding $V_{max}$ for the particulate $P_i$ measured on the goethite surface ($p < 0.05$), indicating that goethite exhibited more of an enzyme-like behavior compared to the other Fe oxides (Fig. 5b).

As a way of normalizing the $V_{max}$ value for each mineral, we performed $P_i$ adsorption experiments to determine the site density for $P_i$ binding on each mineral surface (SI, Fig. S17). While goethite exhibited nearly 3-fold higher capacity for $P_i$ binding than hematite ($26.0 \pm 1.2$ µmol $P_i$ g$^{-1}$ versus $9.1 \pm 0.9$ µmol $P_i$ g$^{-1}$, $p < 0.001$), ferrihydrite had the highest capacity for $P_i$ binding ($184.1 \pm 6.0$ µmol $P_i$ g$^{-1}$) at 7 and 20 times higher than for goethite ($p < 0.001$) and hematite ($p < 0.001$), respectively (SI, Fig. S17). By using the mineral-dependent $P_i$ binding site density to normalize the combined $V_{max}$ for particulate $P_i$ and dissolved $P_i$, we determined the total turnover number for $P_{org}$ dephosphorylation ($k_{cat}$) by each mineral (Fig. 5c). Further highlighting the higher catalytic efficiency of goethite compared to the other Fe oxides, we found that the total $k_{cat}$ for goethite was 2-fold to 4-fold higher than the $k_{cat}$ for hematite ($p < 0.01$) and up to 9-fold greater than the $k_{cat}$ for ferrihydrite ($p < 0.001$) (Fig. 5c). As we have already pointed out, mineral-dependent reactivity did not appear to be due to differences in surface area. Future research on the surface mechanisms underlying this abiotic catalysis will need to address the dependence of the observed difference in catalytic turnover on both the surface chemistry and mineral structure of each mineral.

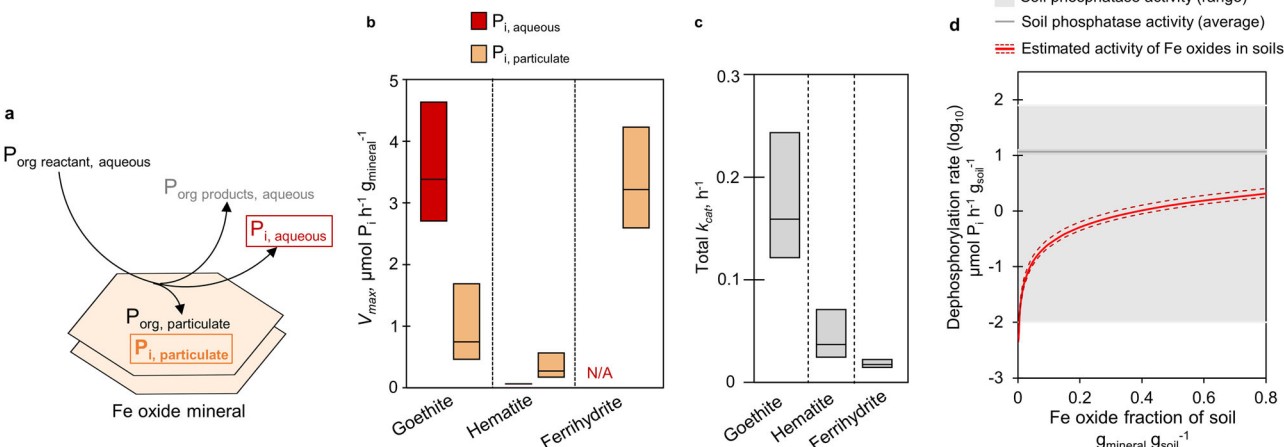

**Fig. 5 | Dephosphorylation kinetics of iron (Fe) oxides and environmental relevance. a** Overview of the different phosphorus (P) species monitored during dephosphorylation reaction of an organic phosphorus ($P_{org}$) reactant with the different Fe oxides; using adenosine triphosphate (ATP) as the $P_{org}$ reactant, we monitored ATP in solution ($P_{org, reactant, aqueous}$), particulate $P_{org}$ species ($P_{org, particulate}$), $P_{org}$ products in solution (adenosine diphosphate, ADP, and adenosine monophosphate, AMP; collectively $P_{org products, aqueous}$), generated inorganic phosphorus ($P_i$) bound to the mineral ($P_{i, particulate}$), and generated $P_i$ in solution ($P_{i, aqueous}$). **b** Apparent maximum rate ($V_{max}$, µmol $P_i$ h$^{-1}$ g$_{mineral}^{-1}$) of $P_{i, aqueous}$ (red) or $P_{i, particulate}$ (light orange) generated during ATP dephosphorylation by goethite, hematite, and ferrihydrite; No $P_{i,aqueous}$ was generated with ferrihydrite (N/A = Not Available). Box plots represent the lower and upper 95% confidence intervals and center value of the $V_{max}$ obtained from the model fit to the experimental kinetics data. **c** Apparent total turnover number (Total $k_{cat}$, h$^{-1}$) of goethite, hematite, and ferrihydrite; each $k_{cat}$ value was calculated by normalizing the sum of the $V_{max}$ values by the density of $P_i$ binding sites (µmol $P_i$ g$_{mineral}^{-1}$) for each mineral (gray). Box plots represent the lower and upper 95% confidence intervals and center value. **d** Rate of Fe oxide-contributed dephosphorylation (µmol $P_i$ h$^{-1}$ g$_{soil}^{-1}$, in log$_{10}$ scale) as a function of increasing Fe oxide fraction in soil, estimated from the combined apparent $V_{max}$ values shown in (**b**). In (**a**) aqueous $P_i$ was determined by UV-vis absorption spectroscopy; particulate $P_i$ was determined by P K-edge X-ray absorption near-edge structure spectroscopy. In (**d**) the dark gray line indicates the average for reported enzymatic dephosphorylation rates, $11.6 \pm 0.8$ µmol $P_i$ h$^{-1}$ g$_{soil}^{-1}$; shown with the gray box is the full range of reported enzymatic rates in soils around the globe reported by Margalef et al., 2017[49]. Source data are provided as a Source Data file.

Here we evaluated the environmental relevance of our findings by comparing reported global values for phosphatase enzyme activity in soils[51] to our Fe oxide-mediated dephosphorylation activity as a function of soil Fe oxide content estimated from our total $V_{max}$ values (Fig. 5d). Even at 2% fraction of Fe oxide content in soil, we found that the estimated rate of Fe oxide-catalyzed dephosphorylation ($0.04-0.06$ µmol $P_i$ h$^{-1}$ g$_{soil}^{-1}$) was above the minimum rate reported for enzymes in soils ($0.01$ µmol $P_i$ h$^{-1}$ g$_{soil}^{-1}$)[51] (Fig. 5d). We further estimated that a soil Fe oxide content of ~20% or greater to be of significance for obtaining abiotic rates that would be approximately within one order of magnitude or less of the averaged value of soil phosphatase rates ($11.6 \pm 0.8$ µmol $P_i$ h$^{-1}$ g$_{soil}^{-1}$)[51] reported in soils globally (Fig. 5d). In accordance with this estimation, we did observe the association of ribonucleotide-sourced $P_i$ primarily with Fe oxides in our heterogeneous soil sample, which had adequate Fe oxide content (~18%) (Figs. 1d and 4g).

## Discussion

The Fe-oxide and Al-bearing minerals are implicated to play a crucial role in the geochemical fate of P due to the strong adsorption of $P_i$ and $P_{org}$ species onto these minerals[3] (Fig. 6). Accordingly, in the soil and sediment samples reacted with a ribonucleotide as a representative $P_{org}$, our data captured the association of the particulate $P_{org}$ fraction derived from the ribonucleotide primarily with Fe oxides and, to a lesser extent, with aluminosilicates (i.e., feldspars, micas, and clays) (Fig. 6). Importantly, with respect to particulate $P_i$ derived from the dephosphorylation of the ribonucleotide, we found that this fraction was associated specifically with Fe and Fe oxides in the sediment and soil matrices (Fig. 6). Experiments with pure phases of the different minerals identified in the natural samples revealed minimal to no dephosphorylation of $P_{org}$ with quartz and the aluminosilicates, which were instead involved only in the adsorption of some of the $P_{org}$ compounds investigated (Fig. 6). In contrast, we obtained exceptional catalytic reactivity (i.e., $P_{org}$

dephosphorylation) with the pure Fe oxides, up to 20-fold greater than with quartz and the aluminosilicates (Fig. 6). We also found that that Fe oxide-catalyzed reaction on the $P_{org}$ compound containing both phosphoanhydride and phosphoester bonds was retained, even in heterogeneous mixtures with quartz or clays. The recycled $P_i$ with pure Fe oxides was found to be largely trapped on the mineral surfaces, consistent with the findings obtained with the natural samples. Thus, we provide evidence of an unaccounted abiotically sourced $P_i$, which would add to residual soil $P_i$ content[52] and $P_i$ inputs from fertilizers[53], that could become subsequently accessible to biota upon desorption from or dissolution of Fe oxides during biotic and abiotic processes[21,54,55] (Fig. 6).

While phosphoanhydride-containing compounds are not commonly detected in soil and sediment systems[36,37], phosphoanhydride bonds are ubiquitous in plant and microbial metabolites, including ATP as the major P content in many microbes[4,56]. We posit that the persistent detection of phosphomonoesters in environmental matrices may be indicative of experimental artifacts during sample extraction or low reactivity of enzymes and minerals towards phosphomonoesters, including those initially derived from dephosphorylating $P_{org}$ with phosphoanhydride bonds. In fact, both enzymatic dephosphorylation[57] and mineral-catalyzed dephosphoryation[16,28] were reported to be higher, by one to two orders of magnitude, for multiphosphorylated compounds than for phosphomonoester compounds. In a similar fashion, goethite and hematite both had minimal reactivity towards several phosphomonoester $P_{org}$ compounds compared to a triphosphorylated $P_{org}$. Here, the observed lack of significant dephosphorylation of phytate and a sugar phosphate by the Fe oxides may contribute to the persistence of these phosphoester $P_{org}$ types in environmental matrices. However, dephosphorylation of a monophosphorylated ribonucleotide by ferrihydrite, was nearly 20-fold greater to that for the triphosphorylated ribonucleotide, with over 80% of the recycled $P_i$ remaining bound to the ferrihydrite. Therefore, both the Fe oxide type and the $P_{org}$ chemistry need to be considered when

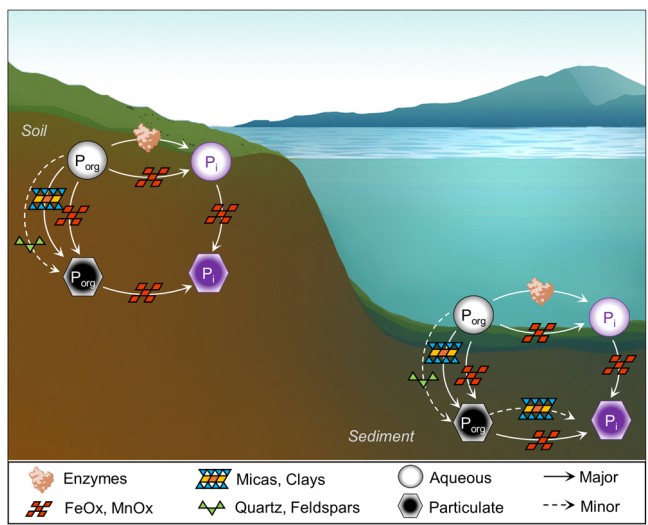

**Fig. 6 | Proposed role of soil and sediment minerals in the geochemical cycling of organic phosphorus ($P_{org}$).** Dissolved $P_{org}$ and inorganic phosphorus ($P_i$) are shown within circles; particulate $P_{org}$ and particulate $P_i$ are shown within hexagons. Enzymatic reaction already included in $P_{org}$ cycling is also shown. Reactions with the following minerals are highlighted: quartz and feldspars (green triangle), micas and clays (blue triangles and yellow/orange squares), and iron and manganese oxides (FeOx and MnOx, dark red squares).

predicting the role of Fe oxides in the extent of the catalytic fate of $P_{org}$ in different environmental matrices.

Our findings corroborate the role of surface chemistry in dictating the extent of catalytic reactivity of the Fe oxide. First, $P_{org}$ dephosphorylation by the Fe oxides was not found to be dependent on surface area. Despite a 14-fold lower surface area for goethite compared to ferrihydrite ($16.0\ m^2\ g^{-1}$ versus $230\ m^2\ g^{-1}$), goethite was found to be the most reactive, dephosphorylating a nearly 7-fold higher amount of the total $P_{org}$ added as ATP. Second, when accounting for the $P_i$ binding site density to normalize the rate of dephosphorylation, goethite still exhibited a turnover number up to 9-fold greater than ferrihydrite. Regarding the relevance of other chemical characteristics of the mineral structure, a study with hematite and goethite proposed Lewis acid sites and Fe coordination to be determining factors for the extent of hydrolytic cleavage of a synthetic $P_{org}$[15,58]. Our results with different naturally-occurring $P_{org}$ compounds showed markedly different reactivity between our three investigated Fe oxides (goethite, ferrihydrite, and hematite) for the same $P_{org}$ compound. This mineral-dependent catalytic reactivity highlights the importance of future inquiries into the roles of surface chemistry, including surface acidity, of the Fe oxides.

Due to the natural abundance of Fe oxides in sediments and soils, it was proposed[16] that the rate of abiotic generation of $P_i$ by Fe oxides may be comparable to that of phosphatase-like enzymes near neutral pH. Here our quantitative analysis provides experimental evidence in support of the proposed contribution of Fe oxides to $P_{org}$ recycling, which would be particularly relevant in Fe oxide-rich matrices with low enzymatic abundance[51] or impeded activity of enzymes immobilized on mineral surfaces[59,60]. While our experiments were performed in the absence of light, light exposure has been shown to enhance ribonucleotide dephosphorylation by some Fe oxides[61]. Therefore, our reported reactivity may be underestimations for Fe oxide reactivity in light-exposed environmental matrices. In sum, our findings highlight an important and yet unaccounted dual catalytic and adsorbent role of Fe oxides that warrants consideration alongside biologically mediated processes in the P cycle. We only considered Fe oxides in this research due to no quantifiable Mn in our lake sediment and forest soil samples, but we acknowledge there can be approximately, on a per-mass basis,

1:10 Mn oxides:Fe oxides in marine sediments[62,63]. It was reported previously that Mn oxides can catalyze dephosphorylation of phosphoanhydride bonds in inorganic polyphosphate[64], phosphoester bonds in a synthetic $P_{org}$[12], and both phosphoester and phosphoanhydride bonds in ribonucleotides and sugar phosphate[16,28]. These previous reports support the proposal that Mn oxides may contribute to the bioavailable P in marine environments. In light of our findings with Fe oxides, we posit that Mn oxides may also participate in catalyzing dephosphorylation of $P_{org}$ and subsequent trapping of the recycled $P_i$, both of which remain to be investigated within the context of Mn oxides in natural samples as shown here for Fe oxides in natural samples.

Current models of P cycling[1,3,65], which include enzymes for $P_{org}$ dephosphorylation and minerals for $P_{org}$ adsorption, do not account for catalytic dephosphorylation of $P_{org}$ by reactive mineral oxides such as Fe oxides and Mn oxides[3,14,16] (Fig. 6). Our proposed redefined role of Fe oxides and related reactive mineral oxides as important catalytic players, if confirmed to be a widespread phenomenon with quantitative significance in different environmental matrices as demonstrated here, would have important implications regarding the addition of an abiotic axiom at the $P_{org}$-$P_i$ nexus in biogeochemical P cycling.

## Methods

### Chemicals
The ribonucleotide compounds [ATP (≥93% purity as determined by our independent LC-MS analysis), ADP (≥95%), and AMP (≥97%)], G6P (≥98%), glucose (≥99.5%) and phytate (≥82% purity as determined by our independent LC-MS analysis) were purchased from MilliporeSigma (St. Louis, MO); adenosine (100%) was obtained from Chem-Impex International (Wood Dale, IL). Phytate dephosphorylation products [inositol pentaphosphate (>98%), inositol tetraphosphate (>98%), inositol triphosphate (>98%), inositol biphosphate (>95%), and inositol monophosphate (>8%)] were purchased from Cayman Chemical (Ann Arbor, MI). Minerals were either purchased or synthesized. Goethite was purchased from Alfa Aesar (Ward Hill, MA), magnetite from MiliporeSigma (St. Louis, MO), hematite from Strem Chemicals (Newburyport, MA), quartz from Honeywell (St. Louis, MO), kaolinite from Fluka (Muskegon, MI), and mica (muscovite) from Spectrum Chemical (Gardena, CA). The illite source clay ISCz-1, obtained from the Clay Minerals Society's repository of Purdue University, was ground in a mortar and pestle before use. Ferrihydrite was synthesized by the method detailed by Schwertmann and Cornell[53]. We obtained XRD spectra using a Bruker D8 Advance powder X-ray diffractometer to confirm the identity of all mineral phases synthesized and purchased matched their reference spectra in the Crystallography Open Database. All other chemicals used were analytical grade and purchased from MilliporeSigma (St. Louis, MO).

### Location and characterization of the natural samples
For natural samples, we obtained samples from a lake sediment and a forest soil. Lake sediment samples were taken from a platform sampling buoy in Missisquoi Bay off Lake Champlain (N44°59'33" W73°8'20") located in Vermont, United States of America. The site, which undergoes seasonal oxic-anoxic cycles, is notable for its high P loading (1.39 mg-P $g^{-1}$ sediment) and presence of redox-sensitive Fe in the sediment (39.8 mg-Fe $g^{-1}$ sediment)[37]. Soil samples were sourced from the Calhoun Critical Zone Observatory (N34°36'33.012" W81°43'40.62"), located in the Sumter National Forest in South Carolina, United States of America. A sample core was taken from the sub soil at a depth of 58–86 cm. The soil is characterized by high Fe oxide content (59.3 mg Fe $g^{-1}$ soil) and minimal amounts of P (0.43 µg P $g^{-1}$ soil)[66]. Both natural samples were sieved to a particle size of 250 µm and ground in a mortar and pestle before use. The pH of the porewater at the Calhoun Critical Zone Observatory soil was determined to be 5.8 at the time of excavation, however the pH has been reported

previously to range between and 4.5 – 6.2[67,68]. The mean annual temperature at the soil sampling site is 289 K (or 16 °C), ranging between 278 K (5 °C) and 298 K (25 °C)[67]. For the Missisquoi Bay sediment sample, the porewater pH was determined to be circumneutral with a temperature of 297 K (24 °C)[69,70]. The natural samples and pure minerals used in this study were characterized by XRF, XRD, and Fe K-edge μ-XRF and μ-XANES. The XRD and XRF data were used in tandem to confirm the presence of specific elements and crystalline phases. The Fe K-edge μ-XRF and μ-XANES data determined the Fe speciation in the natural samples. Location and topography maps with the sample site locations were created using QGIS version 3.24[71] and topography data[72] from the U.S. Geological Survey.

## Elemental XRF analysis of the natural samples

We performed XRF spectroscopy on a Xenemetrix Ex-Calibur EX-2600 spectrometer equipped with a rhodium (Rh) X-ray tube and a silicon (Si) energy-dispersive detector. The samples were placed in a special plastic cup and powder was supported on a 6 μm Mylar film. Standards and samples were used in the form of a pellet and diluted with urea. Data collection was performed at room temperature under vacuum using the nEXt version 1.9 software. The X-ray tube was operated at 20 keV and 10 μA, and fluorescence spectra were collected for 10 min. Both natural samples exhibited significant amounts of Fe, silicon (Si), and Al; the lake sediment also contained potassium (K) (SI, Fig. S1). The XRF analysis did not determine any appreciable amount of manganese in both the lake sediment and forest soil (SI, Fig. S1). Therefore, the presence of manganese oxides was not considered in later analysis.

## Characterization of crystalline minerals in the natural samples via XRD

The XRD diffraction patterns were recorded using a Bruker D8 Advance powder X-ray diffractometer fitted with a copper anode. The diffractometer was operated at 40 kV and 40 mA, with a Göebel Mirror, a Soller 0.2° optic, and a SSD160_2 detector used in 0D mode. Throughout data collection, a temperature of 25 °C was maintained in the sample chamber. The XRD profile was collected in continuous mode with a 2θ step size of 0.005°, a 12 s counting time per step, and from 2θ = 5–72°. Bruker's DIFFRAC.suite V7.5.0 software was used for data collection and subsequent analysis. Peak fitting was performed with DIFFRAC.EVA V6.0.0.7 using candidate peaks in the Crystallography Open Database to determine the percent contribution of each crystalline phase in each natural sample (SI, Fig. S2).

## Characterization of Fe minerals by Fe K-edge μ-XANES and P hotspots by P K-edge μ-XANES in natural samples

Control and ATP-reacted natural samples were both analyzed for Fe species identification by synchrotron analysis. Samples for XANES analysis of ATP-reacted natural samples were prepared in a 500 mL polypropylene bottle by the addition of 400 mg of either the lake sediment and forest soil 400 mL and a 100 μM ATP solution comprised of 0.1 M NaNO₃, 0.01 M NaHCO₃ (adjusted to pH = 7.0 using 2 M HNO₃). Natural samples were reacted for 7 days and stopped by vacuum filtration, then rinsed with a small volume of GenPure water (Thermo Scientific; 18.2 MΩ•cm) to remove excess supernatant. The collected solid was freeze-dried then ground with an agate mortar and pestle. Control dry sediment and soil samples were directly ground with a mortar and pestle. The ground samples were embedded in epoxy (Epotek-301) and thin sectioned on a Buehler PetroThin thin sectioning system to produce 100 mm thick sections mounted on quartz slides.

We performed Fe K-edge and P K-edge μ-XRF mapping and μ-XANES spectroscopy at the Stanford Synchrotron Radiation Lightsource (SSRL) on beamline 2–3. Data was collected using SSRL's MicroEXAFS Data Collector 2.0 software. The energy was selected using a water-cooled Si(111) monochromator. Calibration was performed by setting the maximum of the first derivative of the XANES spectrum of an Fe foil to 7112.0 eV. The beam was focused to a 5 μm spot using a Sigray axially symmetric mirror and fluorescence detection was achieved using a one-element Vortex detector. μ-XRF maps with a step size of 10 microns of the samples were collected at four different energies across the Fe absorption edge: 7122, 7124, 7128 and 7130 eV. Principal component analysis (PCA) was performed on the multi-energy maps to distinguish chemically distinct Fe hot spots within the images, which were processed in Sam's Microprobe Analysis ToolKit (SMAK)[41] version 2.03. These locations were selected for collection of Fe K-edge μ-XANES spectra for the lake sediment and soil sample (SI, Fig. S3; Fig. S4). We also collected μ-XRF maps on this beamline using 3 different energies across the P adsorption edge: 2148, 2152.3, and 2152.5 eV. Because the 2–3 beamline is not as sensitive as the 14-3 beamline, the P mapping was solely used to identify P hotspots in the samples and confirm P localization with Fe. Calibration for P measurements was performed by setting the maximum of the first derivative of the XANES spectrum of GaP to 2152.0 eV.

The Fe K-edge spectra collected at each spot was fit to a set of standards informed by XRF and XRD speciation. For the lake sediment samples: illite (Clay Mineral Society source clay IMt1), and Fe-rich biotite; for the soil samples: hematite; and for both samples: vivianite, ferrihydrite, and goethite. Reference spectra for Fe-rich biotite and vivianite reference XANES spectra were from Sutherland et al. 2020[73] and Hansel et al. 2003[74]. All spectra used for fitting were subjected to the same normalization procedure. The normalization procedure involved fitting a line to the pre-edge region of the spectrum (from −50 – −30 eV relative to E₀) the maximum of the first derivative of the spectra, and a second-order polynomial to the post-edge region (from 80 – 250 eV for lake sediment and from 30 – 250 eV for soil). The pre-edge line was subtracted from the spectrum to remove the background signal and the absorption jump at the first inflection point was set to an intensity of 1.0 based on the difference between the pre- and post-edge fits. Linear combination fitting was performed using the Athena module of the Demeter 0.9.26 IFEFFIT software package[75]. The spectra of different Fe reference spectra were combined to generate a spectrum that fit the spectra of the natural samples. Fits were considered acceptable when visually, the rising edge, white line peak shape, and post-edge region shape matched; the sum of all components was between 0.90–1.10; and if the R-factor was < 0.02. A total of 8 spots were used to generate an average lake sediment Fe speciation and 5 spots used to generate an average soil sample Fe speciation (SI, Fig. S5). Both ferrihydrite and vivianite were identified in the sediment and soil samples (SI, Fig. S5). After determination of Fe speciation, the contribution of each non-crystalline phase (ferrihydrite and vivianite) to the total mineral makeup was calculated from the XRD quantitative analysis of a crystalline Fe-containing mineral, resulting in a total speciation of each natural sample (shown in main text, Fig. 1c).

## Characterization of Al coordination in natural samples by ²⁷Al NMR

Solid state ²⁷Al magic-angle spinning NMR data were collected at room temperature on a Bruker Avance III 400 MHz spectrometer equipped with a 4 mm HX probe using the TopSpin version 3.6.5 software. A zirconia blind-bore rotor with 4 mm diameter was spun at 10 kHz. The resonance frequency of ²⁷Al was 399.73 MHz. ²⁷Al NMR spectra of the lake sediment and soil sample were measured using a recycle delay of 5 s. To achieve an adequate signal-to-noise ratio, a total of 64 scans were collected for the sediment sample and 16 scans were collected for the soil sample. The ²⁷Al chemical shifts were referenced to AlCl₃•6H₂O set to 0.0 ppm. In accordance with previous studies[41,42], we sought to identify different coordination environments of Al based on the following chemical shifts: Al(VI) - 0 ppm, Al(V) - 35 ppm, and Al(IV) - 60 ppm.

## Dephosphorylation reactions of various $P_{org}$ compounds with natural samples and minerals

Triplicate experiments were conducted for 7 days reactions of ribonucleotides (ATP-P; 150 or 300 µM) with natural samples (lake sediment or forest soil; 1 g L$^{-1}$). In 50 mL polypropylene tubes, 40 mL reaction solution was mixed with the adsorbent. The reaction solution comprised of 0.1 M NaNO$_3$, 0.01 M NaHCO$_3$ (adjusted to pH = 7.0 using 2 M HNO$_3$) and the specified concentrations of ribonucleotides. Here, to compare directly the reactivities of the different minerals present across the two different natural samples, we chose to use the standardized conditions of a reaction solution at pH 7.0, ionic strength set by 0.1 M NaNO$_3$, and buffered by 0.01 M NaHCO$_3$ at 25 °C. We acknowledge that the actual porewater conditions in the natural samples, as highlighted above under detailed descriptions of sample characteristics, would be different from the standardized experimental conditions. Reactions were shaken in the absence of light, in an Eppendorf Innova S44i incubator shaker (25 °C, 150 rpm) for 7 days. Reactions were stopped by filtration through a 0.2 µm filter or were centrifuged at 2325 g for 20 min prior to filtration. An aliquot of the filtrate was analyzed for solution $P_i$ immediately and the remaining filtered sample was frozen at −20 °C until analysis by high-resolution LC-MS.

Using the same procedure as above, triplicate experiments were also conducted for 75 min and 7 days reactions of $P_{org}$ compounds (ATP, AMP, phytate, or G6P; 50 µM) with the different minerals (1 g L$^{-1}$) identified in significant fractions in the natural sample composition: quartz, clays (kaolinite, illite), micas (biotite, glauconite), and Fe oxides (ferrihydrite, goethite, magnetite, and hematite). In addition to these single mineral reactions, we performed reactions with two-mineral mixtures containing goethite (1 g L$^{-1}$) and quartz (4 g L$^{-1}$), kaolinite (4 g L$^{-1}$), or ferrihydrite (1 or 4 g L$^{-1}$). Reactions with clays (illite, kaolinite, and mica) were centrifuged at 2325 g for 20 min prior to filtration and analysis of the aqueous phase components. Samples for P K-edge XANES analysis were prepared as described above.

Kinetic experiments were performed in triplicate with goethite and ferrihydrite over a total of 5 time points: (4 h, 8 h, 12 h, 24 h, and 48 h) and five concentrations (25 µM, 50 µM, 125 µM, 200 µM, and 400 µM). The initial rate of velocity for $P_i$ generation (solution-based and mineral-bound) at each concentration was calculated and applied to the Michaelis-Menten model for enzymatic kinetics using the GraphPad Prism version 9.4.0 software. The maximum velocity of Fe oxide-mediated $P_i$ generation was extracted from model fits and subsequently normalized by the $P_i$ adsorption site density for each mineral to calculate the turnover number of $k_{cat}$.

## Identification and quantification of aqueous dephosphorylation products

Solution $P_i$ concentration was quantified by a phosphomolybdate spectrophotometric method using a Cary 60 UV-Visible spectrophotometer[76] using the Cary WinUV Simple Reads version 5.1.3.1042 software. We ascertained that the colorimetric method measured $P_i$ and not the $P_{org}$ products over the course of these experiments. Solution $P_{org}$ compounds (ATP, ADP, AMP; referred in group as AXP moving forward) and adenosine were quantified using a previously established LC-MS method[14]. The LC-MS system used comprised of a Dionex Ultimate 3000 LC coupled to a Q-Exactive mass spectrometer. Measurements and analysis were performed using the Xcalibur version 4.1 software (Thermo Scientific). Compounds were separated using a Waters Acquity UPLC® BEH C18 column (2.1 mm × 100 mm x 1.7 µm) and an 11-min gradient of Solvent A (3% v/v methanol/15 mM acetic acid/10 mM tributylamine) and Solvent B (100% methanol). The mass spectrometer was run in negative ion mode at full MS range (m/z 100 – 600) with a resolution of 70,000. Standards ranging from 0.5 to 15 µM AXP compounds and adenosine were used for calibration. Samples with compound concentrations above 15 µM

were diluted to ensure measurements were within the calibration range. Quality controls were run every 6–9 samples and remained within 30% error. Compounds were quantified using peak area integration from the ion chromatograms.

## Mineral surface-localized P speciation

We used P K-edge bulk XANES spectroscopy to determine the characterize and quantify the particulate $P_{org}$ and $P_i$ following the aforementioned $P_{org}$ compound reactions with the natural samples or the representative minerals. The P K-edge XANES spectroscopy for ribonucleotide reactions was conducted at the Stanford Synchrotron Radiation Lightsource (SSRL) on beamline 14–3. Powdered samples were applied to Mylar tape in a thin layer. The spectra were collected in fluorescence mode with a Vortex detector at room temperature in a helium atmosphere. The vertical slits were set to 1 mm and the horizontal slits to 2.5 mm. A Si(111) monochromator was used to select the energy and calibrated to the energy of the first pre-edge feature of PPh$_4$-Br (2146.96 eV) and there was no noticeable shift 12 and 24 h after the initial calibration. Due to shutdown of the SSRL for maintenance, the P K-edge XANES spectroscopy for non-ribonucleotide reactions (phytate and G6P) was conducted at the MEX2 Beamline of the Australian Synchrotron. Powdered samples were applied to carbon tape in a thin layer. The spectra were collected in fluorescence mode with a Vortex-ME4 detector at room temperature under vacuum. The vertical slits were set to 3 mm and the horizontal slits to 5 mm. A Si(111) monochromator was used to select the energy and calibrated to the energy of the first pre-edge feature of PPh$_4$Br (2146.96 eV). The resulting spectra were analyzed by LCF with the Athena module[75] of the Demeter version 0.9.26 IFEFFIT software package. The mineral associated-P reference spectra were combined to generate a spectrum that fit the spectra of the natural samples. Fits were considered acceptable when visually, the rising edge, white line peak shape, and post-edge region shape matched; the sum of all components was between 0.90 and 1.10; and if the R-factor was < 0.02. To achieve the LCF, we obtained an extensive set of XANES reference spectra for the different minerals (goethite, hematite, ferrihydrite, quartz, kaolinite, illite) by reacting these minerals with $P_i$ or with $P_{org}$ compounds (ATP or AMP). The reference XANES spectra used for the phosphate minerals, vivianite and fluoroapatite, were from Gustafsson et al. 2020[77].

## Statistical analysis

Experimental data points from LC-MS are presented as bar graphs and error bars are calculated as one standard deviation by assuming a standard normal distribution. Values determined from multiple post-processing of experimental data are shown as Box plots generated using GraphPad Prism 9.4.0. Error bars on fractions obtained from LCF modeling of XANES data represent errors of model fits. All statistical significance was calculated using the unpaired $t$-test, except for comparisons of $V_{max}$, $k_{cat}$, and $P_i$ site density values which utilized the $F$-test. Model fitting parameters and associated statistical analysis were performed in GraphPad Prism 9.4.0 and are listed in SI Table S9 and Table S10.

## Reporting summary

Further information on research design is available in the Nature Portfolio Reporting Summary linked to this article.

## Data availability

The authors declare that the data supporting the findings of this study are available within the manuscript, the supplementary information files, and as Datasets deposited in the Oak Ridge National Lab Distributed Active Archive Center for Biogeochemical Dynamics (ORNL DAAC) depository under accession code: https://doi.org/10.13139/ORNLNCCS/2221769. Source data are provided with this paper.

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

## Acknowledgements

This research was funded by the U.S. Department of Energy (Geosciences Program, DE-SC0021172) awarded to L.A. Postdoctoral fellowship for W.T. was provided by Schmidt Science Fellows Program. This work made use of the Integrated Molecular Structure Education and Research Center (IMSERC) facility at Northwestern University, which is supported by the Soft and Hybrid Nanotechnology Experimental (SHyNE) Resource (National Science Foundation, ECCS-2025633) and Northwestern University. The authors thank Dr. Christos Malliakas (Northwestern University) and Nathaniel Barker (Northwestern University) for their assistance with the XRF measurements, Dr. Yuyang Wu for his assistance with the [27]Al NMR measurements, and Edwin Saavedra Cifuentes for assistance with XRF image alignment.

## Author contributions

L.A. conceived and supervised the research. J.J.B. and L.A. designed the research. J.J.B., A.R.K. and W.T. carried out laboratory experiments with natural samples and pure minerals and data analysis. J.J.B., S.E.B., A.R.K. and V.M. performed synchrotron data acquisition and analysis. J.T.S., A.A.T and G.K.D. provided soil and sediment samples. J.J.B., A.R.K and L.A. wrote the manuscript with feedback from all authors.

## Competing interests

The authors declare no competing interests.

## Additional information

**Peer review information** : *Nature Communications* thanks the anonymous reviewers for their contribution to the peer review of this work. A peer review file is available.

