## [Peer Review File · Nature Communications]

Unraveling Iron Oxides as Abiotic Catalysts of Organic Phosphorus Recycling in Soil and Sediment MatricesREVIEWER COMMENTS

Reviewer #1 (Remarks to the Author):

NCOMMS-23-32052: Iron Oxides as Catalytic Traps in Organic Phosphorus Mineralization

Basinski et al., showed and quantified a new and crucial role of iron oxides as an abiotic mineral-enzyme in organic P remineralization. Some critical and interesting roles of environmental Fe and Mn oxides are found in recent research, such as iron- and manganese as catalysis to cause the transformation of simple organic molecules into complex macromolecules (Moore et al., *Nature*, 2023). This manuscript is also an interesting paper to include the quantitative analyses of solution and surface P species transformation into soil and sediment incubation. The author then found that the Fe oxides exhibited up to great high catalytic reactivity (or dephosphorylation activity) for ribonucleotide phosphate mineralization in soils and sediments. Finally, the writings and figure plots are well organized to show the findings and results. However, there still are many concerns that have to be processed before the acceptance of this manuscript. Thus, the current version cannot be accepted by Nature Communications and I suggested a major revision.

Main comments:

- (1) Is ATP (has both C-O-P and P-O-P bonds) a typical organic phosphate in natural environments? Many literatures suggested that, in nature, the main organic P species include phosphate monoester, phosphate diester, and phosphonate (Baldwin, *Environ Chem*, 2013; George et al., *Plant Soil*, 2018). Inositol phosphates are the main organic P storage in terrestrial plants and the most abundant organic P in soils (Turner et al., 2002). How about the catalytic activity of Fe oxides toward the dephosphorylation of phosphate monoester in your soils and sediment incubation? Since they are more abundant Ps in the environment, we cannot neglect their transformation caused by Fe oxides. Wan et al., showed that Fe oxides have low reactivity for C-O-P degradation and high reactivity for P-O-P degradation (Wan et al., *Geochim Cosmochim Acta* 2021; *Sci Total Environ*, 2022). If the catalyzed degradation of ATP by Fe oxides might be specified from the cleavage of P-O-P bond, instead of C-O-P bonds, which will weaken your conclusions about organic P (C-O-P bond-including P) mineralization. You may need to consider adding a C-O-P's organic P to see how Fe oxides catalyze purely organic P mineralization and to strengthen your conclusions.
- (2) How about manganese oxides in organic P mineralization in soils and sediments since Mn oxides also show high reactivity to P transformation (Baldwin et al., *Environ Sci Tech*, 2001; Wan et al. *ACS Earth Space Chem*, 2018).
- (3) You should include the XRF method of P measurement (Figures 2 c-d) in the method section. Did you XRF in the SSRL 2-3 (Energy: 4.9-23 keV)? Could SSRL 2-3 measure P mapping? Why not use SSRL 14-3?
- (4) Line 235-236 Autoclave sterilization? Generally, increasing temperature will significantly change the mineralogy of soils and sediments and might increase the contents of crystalline hematite and goethite? Could you please add the new Fe XANES to compare the mineralogical change of Fe oxides in your soil and sediment samples due to autoclave treatment? Maybe, you also need to consider how this sterilization way affects your findings and conclusions.

(5) What are the parameters in your soil and sediment samples' porewater? Local temperature? You set up pH 7 and 0.1 M NaNO₃ + 0.01 M NaHCO₃ to perform experiments. Why not use the porewater conditions to do experiments? Which might be closer to realistic reaction conditions and show the true catalytic reactivity/rates of Fe oxides in organic P mineralization.

(6) You used XRD to do mineralogical identification. How do you know amorphous phase contents and how do they affect your conclusions?

(7) Figure S6 showed the similar XANES spectra for both mineral-Organic and mineral-Pi? What is the confidentiality of using P K-edge XANES to analyze surface organic and inorganic P?

Overall, the manuscript is of high quality. I assume that considering these concerns/comments might further improve your manuscript quality and consolidate your findings about this new and interesting role of Fe oxides in P biogeochemical cycles in the sub-surface Earth.

References

Baldwin, Darren S. "Organic phosphorus in the aquatic environment." *Environmental Chemistry* 10.6 (2013): 439-454.

Baldwin, Darren S., et al. "Phosphate ester hydrolysis facilitated by mineral phases." *Environmental science & technology* 29.6 (1995): 1706-1709.

George, Timothy S., et al. "Organic phosphorus in the terrestrial environment: a perspective on the state of the art and future priorities." *Plant and Soil* 427 (2018): 191-208.

Moore, Oliver W., et al. "Long-term organic carbon preservation enhanced by iron and manganese." *Nature* (2023): 1-6.

Turner, Benjamin L., et al. "Inositol phosphates in the environment." *Philosophical Transactions of the Royal Society of London. Series B: Biological Sciences* 357.1420 (2002): 449-469.

Wan, Biao, et al. "Rethinking the biotic and abiotic remineralization of complex phosphate molecules in soils and sediments." *Science of The Total Environment* 833 (2022): 155187.

Wan, Biao, et al. "Iron oxides catalyze the hydrolysis of polyphosphate and precipitation of calcium phosphate minerals." *Geochimica et Cosmochimica Acta* 305 (2021): 49-65.

Wan, Biao, et al. "Manganese oxide catalyzed hydrolysis of polyphosphates." *ACS Earth and Space Chemistry* 3.11 (2019): 2623-2634.

Reviewer #2 (Remarks to the Author):

I have concluded that the research described is sufficient quality and potential impact to be accepted for publication in *Nature Communications*, with some minor revision (see below). The manuscript is very well written and presented, and the findings of this study clearly demonstrated that significant mineralization of model organic phosphorus compounds occurred on the surface of iron oxide, which represents a potentially important process that contributes to phosphorus transformations and bioavailability in sediments and soils.

I only have one significant comment/request to make on the manuscript. I am a soil scientist with very little experience or expertise in sediment phosphorus dynamics. With specific regard to the dynamics of

organic phosphorus in soil, I was surprised to note the complete absence of any mention or consideration of the role and function of “iron oxide surface mineralization” in the cycling and bioavailability of inositol phosphates. Inositol phosphates (principally inositol hexaphosphate [commonly referred to as “phytate”]) are widely acknowledged to be important constituents of soil organic phosphorus, and have been shown to be strongly adsorbed on soil colloid surfaces. I would request that the authors consider including some discussion of the potential role of iron oxide surface mineralization in the fate and bioavailability of inositol phosphates in soil.

REVIEWER COMMENTS

Reviewer #1 (Remarks to the Author):

NCOMMS-23-32052: Iron Oxides as Catalytic Traps in Organic Phosphorus Mineralization

Basinski et al., showed and quantified a new and crucial role of iron oxides as an abiotic mineral-enzyme in organic P remineralization. Some critical and interesting roles of environmental Fe and Mn oxides are found in recent research, such as iron- and manganese as catalysis to cause the transformation of simple organic molecules into complex macromolecules (Moore et al., Nature, 2023). This manuscript is also an interesting paper to include the quantitative analyses of solution and surface P species transformation into soil and sediment incubation. The author then found that the Fe oxides exhibited up to great high catalytic reactivity (or dephosphorylation activity) for ribonucleotide phosphate mineralization in soils and sediments. Finally, the writings and figure plots are well organized to show the findings and results. However, there still are many concerns that have to be processed before the acceptance of this manuscript. Thus, the current version cannot be accepted by Nature Communications and I suggested a major revision.

Response: Thank you for acknowledging the new underexplored yet critical role of iron oxides with enzyme-like characteristics in the phosphorus cycle. We appreciate the highlight of our well-written and well-organized by the reviewers. Below we provide details on how we have invested in addressing all the concerns and feedbacks by the reviewer. Notably, we were able to secure additional beamtime at the Australian Synchrotron on short notice to conduct to analyze samples stemming from the additional experiments the author requested with other organic P types. This would have been an impossible task for us to achieve due to the shutdown of the Stanford synchrotron. We are grateful to our diverse team of co-authors that includes beamline scientists from different synchrotron facilities.

Main comments:

Comment 1:

Is ATP (has both C-O-P and P-O-P bonds) a typical organic phosphate in natural environments? Many literatures suggested that, in nature, the main organic P species include phosphate monoester, phosphate diester, and phosphonate (Baldwin, Environ Chem, 2013; George et al., Plant Soil, 2018). Inositol phosphates are the main organic P storage in terrestrial plants and the most abundant organic P in soils (Turner et al., 2002). How about the catalytic activity of Fe oxides toward the dephosphorylation of phosphate monoester in your soils and sediment incubation? Since they are more abundant Ps in the environment, we cannot neglect their transformation caused by Fe oxides. Wan et al., showed that Fe oxides have low reactivity for C-O-P degradation and high reactivity for P-O-P degradation (Wan et al., Geochim Cosmochim Acta 2021; Sci Total Environ, 2022). If the catalyzed degradation of ATP by Fe oxides might be specified from the cleavage of P-O-P bond, instead of C-O-P bonds, which will weaken your conclusions about organic P (C-O-P

bond-including P) mineralization. You may need to consider adding a C-O-P's organic P to see how Fe oxides catalyze purely organic P mineralization and to strengthen your conclusions.

Response: We agree with the reviewer that phosphomonoesters (with C-O-P) are found to be the forms of P_{org} mostly detected in environmental matrices. In our original manuscript, we did present data with both AMP (a phosphomonoester) and ATP reacted with the different Fe oxides, quartz, aluminosilicates. We used the data with the two different ribonucleotides to make a statement that the Fe oxide reactivity would be dependent on the P_{org} types because while goethite and hematite has more reactivity towards ATP than AMP, ferrihydrite had more catalytic reactivity for AMP than for ATP. However, as pointed out by the reviewer, we acknowledge there are different forms of P_{org} compounds with phosphomonoester bonds. To address the concern by the reviewer that we need to consider how different phosphomonoesters may react with the Fe oxides, we conducted additional experiments with a sugar phosphate (glucose-6-phosphate) and phytate (the primary storage in plants), to add to our data with AMP. These new data are illustrated in **Figures 2d and 2e**. We expanded the text to discuss the new results along with the discussion of AMP (**see Lines 212-251**). It is important to acknowledge that P_{org} with phosphoanhydride bonds (P-O-P) are ubiquitous in plant and microbial metabolism and biomass. Therefore, the persistent detection of phosphomonoesters may be indicative of both low reactivity of phosphomonoesters and the accumulation of dephosphorylation products from multiphosphorylated P_{org} compounds containing phosphoanhydride bonds. In fact, in agreement with previous enzymatic dephosphorylation data, we found two of Fe oxides had relatively lower reactivity for phosphomonoesters had lower reactivity compared to ATP, but, contrary to the enzyme results, AMP reactivity was nearly 20-fold greater on ferrihydrite relative to ATP reactivity on ferrihydrite. We added to the discussion section to highlight these findings (**see Lines 407-424**). This remarkable difference in mineral reactivity was only captured by surface characterization by P-XANES. We note that the publication by Wan *et al.* 2022 only monitored the generation of P_i in solution. With regards to experiments with the natural samples, we only performed reactions with ATP due to insufficient amounts of natural samples that precluded us from conducting experiments with both ATP and the entire set of phosphomonoesters.

Lines 212-251:

Amongst the chemical diversity of P_{org} types found in biomolecules, phosphomonoesters are widely found in soils^{34–36,41,42}. To probe Fe oxide reactivity with these other types of P_{org} , we performed reactions involving each of the three Fe oxides (ferrihydrite, hematite, and goethite) with three phosphomonoesters: a ribonucleotide, AMP (AMP-P, 50 $\mu\text{M P}$ or 1.5 mg P L^{-1}); a sugar phosphate, G6P (G6P-P, 50 $\mu\text{M P}$ or 1.5 mg P L^{-1}); and, the primary P storage in terrestrial plants³⁶, inositol hexaphosphate or phytate (phytate-P, 300 $\mu\text{M P}$ or 9.0 mg P L^{-1} (**Fig. 2c, 2d, and 2e**). To compare the Fe oxide reactivity with the other mineral types, we also performed experiments of the phosphomonoester P_{org} compounds reacted with quartz and aluminosilicates (mica, kaolinite, illite) (**Fig. 2c, 2d, and 2e**).

In contrast to the ATP reactions, all the P_i derived from the reacted AMP was retained as particulate P_i while aqueous P_i was absent, a significant finding that was made possible here due to the application of the XANES technique (**Fig. 2c**). On the one hand, the catalytic reactivity of ferrihydrite was higher for AMP than for ATP, as reflected by the 20-fold increase in particulate P_i fraction ($p < 0.001$) (**Fig. 2a and 2c**). On the other hand, the catalytic reactivity of goethite was less for AMP than for ATP as characterized by a 3-fold decrease in particulate P_i fraction ($p < 0.01$) accompanied by no change in the particulate P_{org} fraction ($p = 0.29$) and no measured solution P_i ;

hematite did not display any adsorption or catalytic reactivity towards AMP (**Fig. 2a** and **2c**). Interestingly, in our aforementioned experiments with ATP and goethite, 20–24% of the initial ATP-P remained as AMP-P in solution and no ADP was detected after nearly all the ATP reacted with goethite was transformed or adsorbed (SI, **Fig. S10**). Thus, this accumulation of AMP in the ATP-goethite experiment can be explained by the lower reactivity of goethite for AMP relative to ATP (**Fig. 2c**; SI, **Fig. S10**). When ATP was reacted with hematite, we measured a conversion of 17–23% of the initial ATP-P to ADP-P and 5–7% to AMP-P, consistent with lower reactivity for both ADP and AMP compared to ATP (SI, **Fig. S10**). Notably, the silicate and aluminosilicate minerals did not exhibit any adsorption or catalytic reactivity towards AMP (**Fig. 2c**).

In contrast to AMP, all the silicate and aluminosilicate minerals adsorbed phytate (from 5% to 50% of the total reacted phytate-P) and, rather than an Fe oxide, the clay illite adsorbed the most phytate (**Fig. 2d**). However, similar to the results with AMP and ATP, catalytic reactivity towards phytate was only obtained with Fe oxides, specifically hematite and goethite (**Fig. 2d**). Relative to controls, phytate-derived particulate P_i was higher by 8 – 14% ($p < 0.01$) with hematite and by 24 – 32% with goethite ($p < 0.001$); the particulate P_i with ferrihydrite, however, corresponded to the adsorption of solution P_i in the control experiment (**Fig. 2d**). For G6P, some G6P adsorption was observed with two of the aluminosilicates (mica and kaolinite) and two of the Fe oxides (ferrihydrite and goethite) (**Fig. 2e**). But, as with phytate, the most significant adsorption was with ferrihydrite, with 47 – 78% of the reacted G6P found as particulate P_{org} (**Fig. 2e**). None of the investigated minerals catalyzed G6P dephosphorylation (**Fig. 2e**).

In sum, our data revealed that the silicates and aluminosilicates either had minimal to no catalytic reactivity or exhibited some extent of adsorption reactivity. Only the Fe oxides were found to catalyze the dephosphorylation of both phosphomonoester and phosphoanhydride-bearing compounds. Importantly, the Fe oxide-catalyzed reactions seemed to be dependent on both the mineral surface chemistry and the type of P_{org} species. As proposed previously¹⁴, we expect the differences in reactivity may stem from the binding conformations of different P_{org} on the mineral surface.

Lines 407-424:

While phosphoanhydride-containing compounds are not commonly detected in soil and sediment systems^{34,35}, phosphoanhydride bonds are ubiquitous in plant and microbial metabolites, including ATP as the major P content in many microbes^{4,52}. We posit that the persistent detection of phosphomonoesters in environmental matrices may be indicative of both low reactivity of enzymes and minerals towards phosphomonoesters and the accumulation of remnants of dephosphorylated P_{org} compounds containing phosphoanhydride bonds. Here we show that the Fe oxide-catalyzed reaction for a P_{org} compound containing both phosphoanhydride and phosphoester bonds was retained, even in heterogeneous mixtures with quartz or clays. We also found accumulation of a monophosphorylated ribonucleotide after Fe oxide-catalyzed reactions with a triphosphorylated ribonucleotide. A study on enzymatic dephosphorylation⁵³ reported higher reactivity, by approximately 2 orders of magnitude, for multiphosphorylated compounds than for phosphomonoester compounds. Goethite and hematite, both of which represented the most reactive Fe-oxides, had minimal reactivity towards phosphomonoester P_{org} compounds. This lack of significant dephosphorylation of phytate and a sugar phosphate by the Fe oxide minerals may contribute to the persistence of these phosphoester P_{org} types in environmental matrices. However, dephosphorylation of a monophosphorylated ribonucleotide by ferrihydrite, was nearly 20-fold greater to that for the triphosphorylated ribonucleotide. Therefore, both the Fe oxide type and the P_{org} chemistry need to be considered when predicting the extent of the catalytic fate of P_{org} in different environmental matrices.

Comment 2:

How about manganese oxides in organic P mineralization in soils and sediments since Mn oxides also show high reactivity to P transformation (Baldwin et al., Environ Sci Tech, 2001; Wan et al. ACS Earth Space Chem, 2018).

Response: We agree with the reviewer that Mn oxides are known to be reactive towards organic P compounds, as we had pointed out in the Introduction of our original manuscript (**see Lines 57 and lines 63**). However, the scope of our project is on Fe oxides, especially since there was no quantifiable Mn amount (XRF data) or identifiable Mn-bearing minerals (XRD data) in our natural soil and sediment samples (**See lines 136-137**). We also expanded our discussion to point to the reason why reactivity with Mn oxides was not considered in our experiments but also highlight their possible significance in marine sediments. (**See lines 430-439**).

Lines 136-137:

No manganese-bearing minerals were included in the mineral composition analysis because neither environmental sample exhibited a quantifiable amount of manganese (SI, **Fig. S1**).

Lines 430-439:

Current models of P cycling^{1,3,56}, which include enzymes for P_{org} dephosphorylation and minerals for P_{org} adsorption, do not account for reported catalytic dephosphorylation of P_{org} by Fe oxides and Mn oxides^{3,14,16}. We only consider Fe oxides in this research due to no quantifiable Mn in our forest soil and lake sediment samples, but there can be approximately, on a per-mass basis, 10:1 Fe oxides:Mn oxides in marine sediments^{57,58}. Our quantitative findings imply an important and yet unaccounted dual catalytic and adsorbent role of Fe oxides which warrant consideration alongside biologically mediated processes in the P cycle. Our proposed redefined role of Fe oxides as important catalytic players, if confirmed to be a widespread phenomenon with quantitative significance, would have important implications regarding the addition of an abiotic axiom at the P_{org}-P_i nexus in the biogeochemical P cycling.

Comment 3:

You should include the XRF method of P measurement (Figures 2 c-d) in the method section. Did you XRF in the SSRL 2-3 (Energy: 4.9-23 keV)? Could SSRL 2-3 measure P mapping? Why not use SSRL 14-3?

Response:

As the reviewer pointed out, we did perform the XRF mapping on SSRL's beamline 2-3 and have now included the methodology of those P measurements (**See Lines 517-521**). The P mapping acquired at the 2-3 beamline was solely used for P and Fe correlation analysis. We note that, while the 2-3 beamline at SSRL can perform P mapping, it does not have the same sensitivity as the 14-3 beamline. We also ran the samples with the intention of P mapping at the 14-3 beamline, but we unfortunately do not have the data due to an issue with the memory during data collection. However, because we only used the data from the 2-3 beamline for correlation analysis, the lower sensitivity at that beamline was not an issue.

Lines 517-521:

We also collected μ -XRF maps on this beamline using 3 different energies across the P adsorption edge: 2148, 2152.3, and 2152.5 eV. Because the 2-3 beamline is not as sensitive as the 14-3 beamline, the P mapping was solely used to identify P hotspots in the samples and confirm P

localization with Fe. Calibration for P measurements was performed by setting the maximum of the first derivative of the XANES spectrum of GaP to 2152.0 eV.

Comment 4:

Line 235-236 Autoclave sterilization? Generally, increasing temperature will significantly change the mineralogy of soils and sediments and might increase the contents of crystalline hematite and goethite? Could you please add the new Fe XANES to compare the mineralogical change of Fe oxides in your soil and sediment samples due to autoclave treatment? Maybe, you also need to consider how this sterilization way affects your findings and conclusions.

Response: As the reviewer pointed out, autoclaving of samples containing Fe oxides would likely change the mineralogy of these minerals. For that reason, our natural samples were not autoclaved and biological dephosphorylation was assumed to be minimal due to long-term storage and low carbon loading. We apologize for the typo and have edited the section that included this information (**see Lines 278-286**).

Lines 278-286:

Biologically-mediated P_{org} dephosphorylation was not expected to be significant in our natural samples due to long-term storage (~4 years) of both samples and their low carbon loading (<0.2% g C g^{-1} soil) particularly for the soil sample. Nevertheless, we tested the possibility of residual microbial or enzymatic reactions in the natural samples by performing experiments with an antimicrobial agent or an enzyme denaturing agent, respectively (SI, **Fig. S12**). We determined that these biotic reactions accounted only for 0 – 5% and 23 – 26% of the total reactivity in the sediment and soil samples, respectively (**Fig. 3a** and **2c**). Taken collectively, our findings bring attention to the occurrence of a pool of abiotically generated particulate P_i from mineral-mediated P_{org} transformation that has been hitherto unaccounted for in environmental matrices.

Comment 5:

What are the parameters in your soil and sediment samples' porewater? Local temperature? You set up pH 7 and 0.1 M $NaNO_3$ + 0.01 M $NaHCO_3$ to perform experiments. Why not use the porewater conditions to do experiments? Which might be closer to realistic reaction conditions and show the true catalytic reactivity/rates of Fe oxides in organic P mineralization.

Response: The pH of the porewater at the Calhoun Critical Zone Observatory soil was determined to be 5.8 at the time of excavation, however the reported pH at this site has ranged from 4.5 to 6.2 (Richter *et. al.* 1994; Chen *et. al.* 2020). The mean annual temperature at the soil sampling site is 16 °C with a range of 5 °C to 25 °C (Richter *et. al.* 1994). For the Missisquoi Bay sediment sample, the porewater was determined to be at a temperature of 24 °C with a circumneutral pH (Smith *et. al.* 2011 and Cai *et. al.* 2010). We have included this information in **Lines 468-473**. Rather than using reaction conditions that directly mirrored those of the porewater conditions of the natural systems, we chose to use the standardized conditions of pH

7 and background electrolyte 0.1 M NaNO₃ buffered with 0.01 M NaHCO₃ at 25 °C to be able to compare directly the reactivities of the different minerals present across the two different systems. Please see **lines 548-553** in the main text for the inclusion of this reasoning.

Lines 468-473:

The pH of the porewater at the Calhoun Critical Zone Observatory soil was determined to be 5.8 at the time of excavation, however the pH has been reported previously to range between and 4.5 to 6.2^{57,58}. The mean annual temperature at the soil sampling site is 289 K (or 16 °C), ranging temperatures found to range between 278 K (5 °C) and 298 K (25 °C)⁵⁷. For the Missisquoi Bay sediment sample, the porewater pH was determined to be circumneutral with a temperature of 297K (24.0 °C)^{59,60}.

Lines 548-553:

Here, to compare directly the reactivities of the different minerals present across the two different natural samples, we chose to use the standardized conditions of a reaction solution at pH 7.0, ionic strength set by 0.1 M NaNO₃, and buffered by 0.01M NaHCO₃ at 298K (25 °C). We acknowledge that the actual porewater conditions in the natural samples, as highlighted above under detailed descriptions of sample characteristics, would be different from the standardized experimental conditions.

Comment 6:

You used XRD to do mineralogical identification. How do you know amorphous phase contents and how do they affect your conclusions?

Response: We acknowledge that our characterization method focused on the crystalline phases via XRD and that only the amorphous phases were that of Fe phases characterized via XANES. The scope of our technical analysis did not account for possible amorphous silicate and aluminosilicate phases. However, according to our data, these silicate-bearing minerals all had low catalytic and adsorption reactivities compared to the Fe oxides for our representative P_{org}. Therefore, we do not expect the presence of amorphous phases would affect the fate of the P_{org} reactant. The main text has been modified (**See Lines 151 – 157**) to express this caveat.

Lines 151-157:

Moreover, our XRD and Fe XANES data would not account for the possible presence of low-crystallinity silicate or aluminosilicate phases in the natural samples. However, as will be discussed in the next section, there was minimal to no catalytic reactivity of silicate or aluminosilicate silicates towards the different P_{org} compounds (**Fig. 2**). Consistently, based on our analysis, we found that about 80% (or more) on a per-mass basis of both sediment and soil samples was comprised of silicate minerals of different types (quartz, micas, feldspars, clays), and less than 20% constituted the Fe oxide fraction (**Fig. 1c**).

Comment 7:

Figure S6 showed the similar XANES spectra for both mineral-Porgnic and mineral-Pi? What is the confidentiality of using P K-edge XANES to analyze surface organic and

inorganic P?

Overall, the manuscript is of high quality. I assume that considering these concerns/comments might further improve your manuscript quality and consolidate your findings about this new and interesting role of Fe oxides in P biogeochemical cycles in the sub-surface Earth.

Response: Recent studies performed by Prietzel *et al.*, 2016, Klein *et al.*, 2019, and Eusterhues *et al.*, 2023, have shown that P K-edge spectra LCF can be used to differentiate the difference between Fe—P_{org} and Fe—P_i. This is due to the analysis of the pre-edge feature, shifting of the white line, and post-edge features. As pointed out by the reviewer, it is difficult to discern differences in the P_i and P_{org} reference spectra and reaction spectra in the original S6 figure. We have included an inset that focuses on the white line region and the post-edge region so this distinction is more apparent. Please see the new Figure S6.

References

- Wan *et al.* Rethinking the biotic and abiotic remineralization of complex phosphate molecules in soils and sediments. *Science of The Total Environment* **833**, 155187 (2022).
- Richter *et al.* Soil chemical change during three decades in an old-field loblolly pine (*Pinus taeda* L.) ecosystem. *Ecology* **75**, 1463 (1994).
- Chen *et al.* Iron-mediated organic matter decomposition in humid soils can counteract protection. *Nature Communications* **11**, 2255 (2020).
- Smith *et al.* Relating sediment phosphorus mobility to seasonal and diel redox fluctuations at the sediment–water interface in a eutrophic freshwater lake. *Limnology and Oceanography* **56**, 2251–2264 (2011).
- Cai *et al.* Carbon Cycling and the Coupling Between Proton and Electron Transfer Reactions in Aquatic Sediments in Lake Champlain. *Aquat Geochem* **16**, 421–446 (2010).
- Prietzel *et al.* Reference spectra of important adsorbed organic and inorganic phosphate binding forms for soil P speciation using synchrotron-based K-edge XANES spectroscopy. *Journal of Synchrotron Radiation* **23**, 532–544 (2016).
- Klein *et al.* Abiotic phosphorus recycling from adsorbed ribonucleotides on a ferrihydrite-type mineral: Probing solution and surface species. *Journal of Colloid and Interface Science* **547**, 171–182 (2019).
- Eusterhues *et al.* Importance of inner-sphere P-O-Fe bonds in natural and synthetic mineral-organic associations. *Science of The Total Environment* **905**, 167232 (2023).

Reviewer #2 (Remarks to the Author):

I have concluded that the research described is sufficient quality and potential impact to be

accepted for publication in Nature Communications, with some minor revision (see below). The manuscript is very well written and presented, and the findings of this study clearly demonstrated that significant mineralization of model organic phosphorus compounds occurred on the surface of iron oxide, which represents a potentially important process that contributes to phosphorus transformations and bioavailability in sediments and soils.

Response: We thank the reviewer for their kind words regarding our effort to present a well-written manuscript and acknowledging the important contribution of our research findings.

Main comments:

Comment 1:

I only have one significant comment/request to make on the manuscript. I am a soil scientist with very little experience or expertise in sediment phosphorus dynamics. With specific regard to the dynamics of organic phosphorus in soil, I was surprised to note the complete absence of any mention or consideration of the role and function of “iron oxide surface mineralization” in the cycling and bioavailability of inositol phosphates. Inositol phosphates (principally inositol hexaphosphate [commonly referred to as “phytate”]) are widely acknowledged to be important constituents of soil organic phosphorus, and have been shown to be strongly adsorbed on soil colloid surfaces. I would request that the authors consider including some discussion of the potential role of iron oxide surface mineralization in the fate and bioavailability of inositol phosphates in soil.

Response: We agree with the reviewer that inositol phosphate or phytate is an important component of soil P_{org} reservoirs. We conducted additional experiments involving reactions of phytate with the three Fe oxides (ferrihydrite, hematite, and goethite), quartz, and the aluminosilicates with inositol hexaphosphate or phytate. We also performed experiments with the sugar phosphate G6P, another important P_{org} in plant and microbial systems. The results can be found in **new Figure 2d, and 2e** and the accompanying discussion in **see Lines 212-251**. Our results show that the catalytic reactivity of Fe oxides (specifically goethite and hematite) towards phytate was significantly lower than with ATP, but the Fe oxide was the only mineral demonstrating catalytic dephosphorylation of phytate compared to the other silicate and aluminosilicate minerals.

Lines 212-251:

Amongst the chemical diversity of P_{org} types found in biomolecules, phosphomonoesters are widely found in soils^{34–36,41,42}. To probe Fe oxide reactivity with these other types of P_{org} , we performed reactions involving each of the three Fe oxides (ferrihydrite, hematite, and goethite) with three phosphomonoesters: a ribonucleotide, AMP (AMP-P, 50 $\mu\text{M P}$ or 1.5 mg P L^{-1}); a sugar phosphate, G6P (G6P-P, 50 $\mu\text{M P}$ or 1.5 mg P L^{-1}); and, the primary P storage in terrestrial plants³⁶, inositol hexaphosphate or phytate (phytate-P, 300 $\mu\text{M P}$ or 9.0 mg P L^{-1} (**Fig. 2c, 2d, and 2e**). To compare the Fe oxide reactivity with the other mineral types, we also performed experiments of the phosphomonoester P_{org} compounds reacted with quartz and aluminosilicates (mica, kaolinite, illite) (**Fig. 2c, 2d, and 2e**).

In contrast to the ATP reactions, all the P_i derived from the reacted AMP was retained as particulate P_i while aqueous P_i was absent, a significant finding that was made possible here due to the application of the XANES technique (**Fig. 2c**). On the one hand, the catalytic reactivity of ferrihydrite was higher for AMP than for ATP, as reflected by the 20-fold increase in particulate P_i fraction ($p < 0.001$) (**Fig. 2a and 2c**). On the other hand, the catalytic reactivity of goethite was less

for AMP than for ATP as characterized by a 3-fold decrease in particulate P_i fraction ($p < 0.01$) accompanied by no change in the particulate P_{org} fraction ($p = 0.29$) and no measured solution P_i ; hematite did not display any adsorption or catalytic reactivity towards AMP (**Fig. 2a** and **2c**). Interestingly, in our aforementioned experiments with ATP and goethite, 20–24% of the initial ATP-P remained as AMP-P in solution and no ADP was detected after nearly all the ATP reacted with goethite was transformed or adsorbed (SI, **Fig. S10**). Thus, this accumulation of AMP in the ATP-goethite experiment can be explained by the lower reactivity of goethite for AMP relative to ATP (**Fig. 2c**; SI, **Fig. S10**). When ATP was reacted with hematite, we measured a conversion of 17–23% of the initial ATP-P to ADP-P and 5–7% to AMP-P, consistent with lower reactivity for both ADP and AMP compared to ATP (SI, **Fig. S10**). Notably, the silicate and aluminosilicate minerals did not exhibit any adsorption or catalytic reactivity towards AMP (**Fig. 2c**).

In contrast to AMP, all the silicate and aluminosilicate minerals adsorbed phytate (from 5% to 50% of the total reacted phytate-P) and, rather than an Fe oxide, the clay illite adsorbed the most phytate (**Fig. 2d**). However, similar to the results with AMP and ATP, catalytic reactivity towards phytate was only obtained with Fe oxides, specifically hematite and goethite (**Fig. 2d**). Relative to controls, phytate-derived particulate P_i was higher by 8 – 14% ($p < 0.01$) with hematite and by 24 – 32% with goethite ($p < 0.001$); the particulate P_i with ferrihydrite, however, corresponded to the adsorption of solution P_i in the control experiment (**Fig. 2d**). For G6P, some G6P adsorption was observed with two of the aluminosilicates (mica and kaolinite) and two of the Fe oxides (ferrihydrite and goethite) (**Fig. 2e**). But, as with phytate, the most significant adsorption was with ferrihydrite, with 47 – 78% of the reacted G6P found as particulate P_{org} (**Fig. 2e**). None of the investigated minerals catalyzed G6P dephosphorylation (**Fig. 2e**).

In sum, our data revealed that the silicates and aluminosilicates either had minimal to no catalytic reactivity or exhibited some extent of adsorption reactivity. Only the Fe oxides were found to catalyze the dephosphorylation of both phosphomonoester and phosphoanhydride-bearing compounds. Importantly, the Fe oxide-catalyzed reactions seemed to be dependent on both the mineral surface chemistry and the type of P_{org} species. As proposed previously¹⁴, we expect the differences in reactivity may stem from the binding conformations of different P_{org} on the mineral surface.

REVIEWER COMMENTS

Reviewer #1 (Remarks to the Author):

I appreciate the authors had processed most of my comments and after major revision, the manuscript looks better than the initial version. I am still very concerned about use P K-edge XANES to distinguish P species within different mineral phases and within different P species. In figure 2c and lines 225-230/423-426, the author observed the significant hydrolysis of AMP by ferrihydrite. But with comparison, ferrihydrite showed super limited hydrolysis ability of phytate (Pi in phytate may come ~4% impurity) and G6P, which three P species all contain only one P-O-C bond. Why here ferrihydrite can hydrolyze more than 70% AMP? Why other two Fe oxides show limited AMP hydrolysis ability? what mechanism/surface structure drives such a high amount of AMP's P-O-C cleavage by ferrihydrite, but not for G6P's P-O-C breakage? Baldwin (1995) indicated no such big difference or P-O-C cleavage between different iron oxides and Tong (2020) indicates that the main force to drive C-O-P breakage comes from the exposure facets of crystalline iron oxides. Thus, this data does not make sense, which may come from P K-edge XANES fitting (70% Pi in particulate phase, but without any release of Pi into solution, which does not also make sense). I suggest the author should check P K-edge XANES for these AMP-iron oxide samples to get accurate fitting results. Thus, I recommend a minor revision and the author should process this important concern, which might significantly challenge the conclusions here.

References

Li, Tong, et al. "Enhanced hydrolysis of p-nitrophenyl phosphate by iron (hydr) oxide nanoparticles: Roles of exposed facets." *Environmental Science & Technology* 54.14 (2020): 8658-8667.

Baldwin, Darren S., et al. "Phosphate ester hydrolysis facilitated by mineral phases." *Environmental science & technology* 29.6 (1995): 1706-1709.

Reviewer #2 (Remarks to the Author):

I have considered the detailed and comprehensive responses by the authors to my original comment and the much more detailed comments made by the other reviewer, and concluded that these adequately addressed our collective questions. Accordingly, I recommend that the manuscript be accepted for publication without any further revision.

REVIEWER COMMENTS

Reviewer #1 (Remarks to the Author):

To address clearly the comments provided by Reviewer 1, we divided up the one paragraph provided by the reviewer into several comments as detailed below.

Comment 1: I appreciate the authors had processed most of my comments and after major revision, the manuscript looks better than the initial version.

Response: We thank the reviewer for acknowledging our efforts. It was a herculean effort to secure on very short notice additional beamtime at a synchrotron in Australia during the shutdown of the Stanford synchrotron in California. We have done experiments with additional P_{org} biomolecules with five different minerals, in addition to the experimental data with two other P_{org} compounds that were already presented in the paper.

Comment 2: I am still very concerned about use P K-edge XANES to distinguish P species within different mineral phases and within different P species.

Response: Based on this comment from the reviewer, we would like to clarify that **we do not apply** the XANES technique beyond its limitation. As pointed out in the Introduction, the P K-edge XANES technique has been demonstrated to distinguish between organic P and inorganic P bound to Fe oxides (see **Lines 72-79**):

“Advances in mineral surface characterization by synchrotron-based P K-edge X-ray absorption near-edge structure (XANES) spectroscopy have made it possible to distinguish between P_i and P_{org} bound to Fe in minerals²⁸ or Fe oxides in a soil matrix²⁹. In terms of monitoring mineral-mediated P_{org} recycling, application of the XANES technique with ferrihydrite revealed the generation of particulate P_i from adsorbed ribonucleotides, while P_i was notably absent from solution¹⁴. This latter finding, which was confirmed by quantifying the dephosphorylated organic products in the solution by LC-MS¹⁴, highlights the need to perform quantitative analysis of particulate P_i, in addition to dissolved P_i, especially for minerals such as Fe oxides with strong adsorption affinity for P_i species.”

Furthermore, we have pointed out in the original version of the manuscript both the powerful insights that can be obtained from the XANES technique and for which analysis this technique can have limitations (see **Lines 326-332**):

“It was not possible to use the P K-edge XANES spectra to distinguish the specific P species associated with the different aluminosilicates nor the specific Fe oxide minerals associated with either particulate P_i or particulate P_{org} species on Fe oxides (SI, **Fig. S14**). Specifically, we were able to employ the LCFs to distinguish P_i associated with Ca using apatite as a reference, P (without discriminating between P_i or P_{org}) associated with aluminosilicates, P_i associated with Fe in Fe oxides, P_{org} species associated with Fe in Fe oxides, and P_i in Fe-P_i clusters using vivianite as reference (**Fig. 4c, 4d, and 4e**).”

Therefore, in response to the statement by the reviewer, we did not apply P K-edge XANES to all P species and all minerals. This XANES technique was used to quantify the fraction of particulate P_i and particulate P_{org}. This can be done for any type of P_{org} types but we can't distinguish between different P_{org} species present simultaneous in the particulate fraction. Importantly, we employed high-resolution mass spectrometry to monitor different organic species (both phosphorylated and dephosphorylated organic species) in solution, from which preliminary quantitation of particulate P_i

can be determined from performing a mass balance on P. For example, in mineral-mediated analysis of ATP, the following mass balance will provide a preliminary account of particulate P_i : $(P_{i, particulate}) = (ADP_{solution}) + 2(AMP_{solution}) - (P_{i, solution})$. Using such mass balance, we have highlighted previously that the absence of P_i in solution does not mean lack of catalyzed reaction; importantly, the presence of dephosphorylated organics highlight missing P_i determined to be particulate P_i (Klein *et. al.*, 2019).

Comment 3: In figure 2c and lines 225-230/423-426, the author observed the significant hydrolysis of AMP by ferrihydrite. But with comparison, ferrihydrite showed super limited hydrolysis ability of phytate (Pi in phytate may come ~4% impurity) and G6P, which three P species all contain only one P-O-C bond. Why here ferrihydrite can hydrolyze more than 70% AMP? Why other two Fe oxides show limited AMP hydrolysis ability? what mechanism/surface structure drives such a high amount of AMP's P-O-C cleavage by ferrihydrite, but not for G6P's P-O-C breakage?

Response: It is reasonable to expect that different energy would be required to cleave the P-O-C bond in our different investigated phosphomonoesters because the different organic structures attached to the phosphoester bond: a ribonucleoside composed of a sugar attached to heterocycling nitrogenous ring in AMP, versus only a sugar moiety in G6P, versus a inositol in phytate. In fact, Wan *et. al.* (2022) reported, based only on monitoring aqueous P_i , 10-fold to 25-fold higher P_i production from AMP than from G6P and phytate reacted with hematite—these previous experiments were done with 60% less mineral (per mass) in solution and with 20-fold higher concentration for G6P and AMP and 3-fold higher concentration for phytate compared to our experiments. Furthermore, different rates of enzymatic dephosphorylation were reported for these different phosphomonoester P_{org} types (Solhtalab *et. al.*, 2021). Therefore, based on these previous findings, it is not surprising that we found less P_i production with G6P compared to AMP reacted with the Fe oxides.

With respect to mechanisms, we would like to point out that the motivation of the current study was to explore whether mineral-mediated catalysis of P_{org} recycling, which was proposed with pure minerals, can actually occur with different natural P_{org} sources, with different minerals, and, importantly, within natural environmental samples obtained from a forest soil and a lake sediment. Now, it is worthwhile for the field to invest in figuring out the mechanisms of this process because we have provided evidence, for the first time, that this mineral-mediated catalysis can occur within environmental samples and, importantly, we obtained abiotic rates that were comparable to enzymatic rates reported in certain soils. Understanding and determining the mechanisms that influence Fe oxide reactivity, which are beyond the scope of the current manuscript, is an important subject for future investigation. We added discussion on exploration of these mechanisms in recently published studies as well as pointing out areas that need to be investigated to gain further insights. (see **Lines 452-463**):

“Our findings corroborate the role of surface chemistry in dictating the extent of catalytic reactivity of the Fe oxide. First, P_{org} dephosphorylation by the Fe oxides was not found to be dependent on surface area. Despite a 14-fold lower surface area for goethite compared to ferrihydrite ($16.0 \text{ m}^2 \text{ g}^{-1}$ versus $230 \text{ m}^2 \text{ g}^{-1}$), goethite was found to be the most reactive, dephosphorylating nearly 7-fold higher amount of the total P_{org} added as ATP. Second, when accounting for the P_i binding site density to normalize the rate of dephosphorylation, goethite still exhibited a turnover number up to 9-fold greater than ferrihydrite. Regarding the relevance of other chemical characteristics of the mineral structure, a study with hematic proposed Lewis acid sites and Fe coordination to be determining factors for the extent of hydrolytic cleavage of a synthetic P_{org} by

this Fe oxide^{15,57}. Our results with naturally-occurring P_{org} showed markedly different reactivity between our three investigated Fe oxides (goethite, ferrihydrite, and hematite) for the same P_{org} compound. Therefore, is Fe oxide-dependent catalytic reactivity highlights the importance of further investigation into the importance of Fe oxide surface chemistry and surface acidity.”

Comment 4: Baldwin (1995) indicated no such big difference or P-O-C cleavage between different iron oxides and Tong (2020) indicates that the main force to drive C-O-P breakage comes from the exposure facets of crystalline iron oxides. Thus, this data does not make sense, which may come from P K-edge XANES fitting (70% Pi in particulate phase, but without any release of Pi into solution, which does not also make sense).

References

Li, Tong, et al. "Enhanced hydrolysis of p-nitrophenyl phosphate by iron (hydr) oxide nanoparticles: Roles of exposed facets." *Environmental Science & Technology* 54.14 (2020): 8658-8667.

Baldwin, Darren S., et al. "Phosphate ester hydrolysis facilitated by mineral phases." *Environmental science & technology* 29.6 (1995): 1706-1709.

Response: It is not clear to us what the objection of the Reviewer is because we are reporting our data that were obtained from meticulously performed experiments with independent replicates. It is worthwhile to note that differences in P_{org} dephosphorylation by different Fe oxides has been reported previously by Li *et. al.* 2020, a reference highlighted by the reviewer (as stated above). Specifically, Li *et. al.* (2020) reported greater dephosphorylation of pNPP, a synthetic P_{org}, by goethite than by hematite. In agreement with this finding, our results showed no reactivity with hematite but there was reactivity with goethite and ferrihydrite, albeit direct comparison between pNPP and the ribonucleotide AMP is not appropriate (as we already discussed above).

As highlighted both in the abstract and the introduction of our manuscript, the motivation for using XANES technique is because it is critical to consider Fe oxide surfaces as both adsorbents and catalysts whereby generated P_i from the mineral catalysis may be trapped on the surface and thus absent from the solution phase. As we have pointed out in the Introduction, a previous study (Klein *et al.*, 2019) had reported the absence of P_i in solution while dephosphorylated organic species were detected in solution by high-resolution LC-MS, which led to the quantification of missing P_i bound to ferrihydrite an Fe oxide surface (See our response to Comment 2 above).

Comment 6: I suggest the author should check P K-edge XANES for these AMP-iron oxide samples to get accurate fitting results. Thus, I recommend a minor revision and the author should process this important concern, which might significantly challenge the conclusions here.

Response: We were transparent with our XANES data and how they were fitted (see SI Figs S7-S10, Fig. S15, and Table S3, S5-S7) in response to comments during the first round of reviews. It is important to note that the application of the XANES technique has been validated previously, as we have highlighted in the Introduction (also see our response to Comment 2). This technique has been previously documented to distinguish between inorganic P and organic P either bound to pure Fe oxides or found in the environmental matrices (Prietzl *et. al.* 2016, Eusterhues *et. al.*, 2023). Here we are applying this technique to monitor, for the first time the *de novo* generation of inorganic P

from organic P reacted with environmental samples and pure forms of Fe oxide and other minerals. To add further clarification on how the XANES data were used, we added 3 additional figure panels to Figure 4 with close-up illustrations of different part of the XANES spectra and we expanded our results section to explain in more details how these spectra were analyzed.

Reviewer #2 (Remarks to the Author):

I have considered the detailed and comprehensive responses by the authors to my original comment and the much more detailed comments made by the other reviewer, and concluded that these adequately addressed our collective questions. Accordingly, I recommend that the manuscript be accepted for publication without any further revision.

Response: We thank the reviewer for their time in the thorough review of our manuscript in both rounds. Thank you for recommending our manuscript for publication.

References:

- Klein, A. R., Bone, S. E., Bakker, E., Chang, Z. & Aristilde, L. Abiotic phosphorus recycling from adsorbed ribonucleotides on a ferrihydrite-type mineral: Probing solution and surface species. *Journal of Colloid and Interface Science* **547**, 171–182 (2019).
- Wan, B., Huang, R., Diaz, J. M. & Tang, Y. Rethinking the biotic and abiotic remineralization of complex phosphate molecules in soil and sediments. *Science of the Total Environment* **833**, (2022).
- Solhtalab, M., Klein, A. R. & Aristilde, L. Hierarchical Reactivity of Enzyme-Mediated Phosphorus Recycling from Organic Mixtures by *Aspergillus niger* Phytase. *J. Agric. Food Chem.* **69**, 2295–2305 (2021).
- Li, T. *et al.* Enhanced Hydrolysis of p-Nitrophenyl Phosphate by Iron (Hydr)oxide Nanoparticles: Roles of Exposed Facets. *Environ. Sci. Technol.* **54**, 8658–8667 (2020).
- Baldwin, D. S., Beattie, J. K., Coleman, L. M. & Jones, D. R. Phosphate Ester Hydrolysis Facilitated by Mineral Phases. *Environ. Sci. Technol.* **29**, 1706–1709 (1995).
- Prietzl, J. *et al.* Reference spectra of important adsorbed organic and inorganic phosphate binding forms for soil P speciation using synchrotron-based K-edge XANES spectroscopy. *Journal of Synchrotron Radiation* **23**, 532–544 (2016).
- Eusterhues, K. *et al.* Importance of inner-sphere P-O-Fe bonds in natural and synthetic mineral-organic associations. *Science of The Total Environment* **905**, 167232 (2023).

REVIEWERS' COMMENTS

Reviewer #1 (Remarks to the Author):

The authors have processed my concerns well and I suggest the current version of this manuscript can be accepted for the publication.